

**Extended Application of the CNOP-P method in the Inner Mongolia using the**

**Common Land Model**

Bo Wang[a,b,d], Zhenhua Huo[c,d], Yujing Yuan[e], Shang Wu[b]

[a] Institute of Applied Mathematics, Henan University, 475004 Kaifeng, China

[b] School of Mathematics and Statistics, Henan University, 475004 Kaifeng, China

[c] University of Chinese Academy of Sciences, 100049, Beijing, China

[d] LASG, Institute of Atmospheric Physics, Chinese Academy of Sciences, 100029, Beijing, China

[e] School of Mathematics, Shandong University, 250100 Jinan, China

*Correspondence to*: Bo Wang (wangbo_sdu@163.com)



**Abstract:** An extension method of the conditional nonlinear optimal perturbation about parameter (CNOP-P) is adopted to study the soil parameter optimization for the Hulunbeier Steppe within the common land model (CoLM) with the differential evolution (DE) method. Using National Center for Environmental Prediction/Department of Energy (NCEP/DOE) Atmospheric Model Intercomparison Project-Ⅱ (AMIP-Ⅱ) 6-hourly Reanalysis Gaussian Grid data and National Meteorological Center (NMC) Reanalysis 6-hourly surface fluxes data, three experiments (Ⅰ and Ⅱ) were designed to study the impact of the percentages of sand and clay of the shallow soil in CoLM on simulating the shallow soil moisture. To study the shallow soil moisture and the latent heat flux simultaneously, experiment (Ⅲ) is designed. The optimal parameters obtained by the extended CNOP-P method are used to predict the shallow soil moisture in the following month. In all the three experiments, after optimization stage, the optimal soil parameters could significantly improve the simulation ability of CoLM in the Inner Mongolia to the shallow soil moisture at the stage of prediction; the optimal parameters attained by the double-parameter optimal experiment could make CoLM simulate the shallow soil moisture better than the single-parameter optimal experiment in the optimization slot. Moreover, the results of experiments (Ⅰ and Ⅱ) justify the conclusion that the more accurate the atmospheric forcing data and observation data are, the more effective the results of optimization will be.

*Keywords:* CNOP-P, parameter optimization, shallow soil moisture, CoLM, Inner Mongolia

## 1. Introduction

With the population explosion, the atmospheric environment pollution, the marine ecological deterioration, the land erosion and desertification, the sharp drop in fores resources, the acid harm, the extinction of species, the water pollution and the toxic waste pollution, global warming has recently been a serious problem that more and more scientists are concerned with. Global worming will result in the global climate



change including glacial ablation, sea level rise, floods, landslides, debris flow and so
on. In addition, with global warming, the atmospheric temperature increases and the
evaporation increases, so there is a severer trial in the areas which usually lack of rain
with arid climate, the semi-arid areas which were not covered with plants well may be
degenerated into the semi-desert areas, and so global warming may induce the
acceleration of the desertification in the inland areas.
China is one of the countries which are severely affected by desertification.
Desertification in Northern China has been an important problem which needs to be
solved urgently for national economic and social development, with the wide
distribution and rapid development of the desertification. In China, Inner Mongolia
Autonomous Region is the most serious region affected by the desertification , and in
recent years, there are great temperature and precipitation changes(Su et al., 2008;
Han et al., 2010; Zhang et al., 2014), which can influence the climate of Inner
Mongolia. So, it is necessary to predict accurately the variables, such as temperature,
precipitation, soil moisture and latent heat et al., which are very important to study the
drought degree of Inner Mongolia Autonomous Region so that it could supply some
help for the management and control of desertification, especially for the agricultural
and animal husbandry production.
Hulunbeier Steppe is located in the northeast grassland of China's Inner Mongolia
Autonomous Region, west of the Greater Khingan Mountains. The Greater Khingan
Mountains separates the Hulunbuir Steppe into two kinds of climate. The eastern
ridge is the monsoon climate and the western ridge is the continental climate from the
point of the climate types. From the point of the annual precipitation types, the eastern
ridge is the semi-humid climate and the western ridge is a semi-arid climate. The
special geographical position of Hulunbuir brings about that the total characteristics in
climate of Hulunbiur Steppe is cold and dry in winter, hot and rainy in summer and
the annual and daily temperature differences are large. Since 1999, the climate of
Hulunbiur became unusual: the annual precipitation is low, the spring is droughty and
windy, the summer continues the high temperature, water evaporation capacity
increases and the level of drought increases. Located in the arid and semi-arid area of



North China, the Hulunbiur Steppe is one of the important livestock husbandry bases
and high quality natural pastures in China, so, it is extremely important to study the
land surface of the Hulunbiur Steppe.
The land surface processes mainly study all the processes which are closely related
to the atmosphere movement at the underlying surface. They contain the complex soil
water heat transmission and vegetation physiological and biochemical processes.
They are one of the fundamental biochemical and physical processes which could
affect atmospheric circulation and climatic change. In order to understand the land
surface processes better, it is worth improving the land surface models (LSMs). Sun
(2005) pointed out that the parameter values in some physical processes are inaccurate,
and the simulations of LSMs are also inaccurate with the impact of inaccurate
parameter values. So, it is meaningful to optimize the parameters of LSMs. Some
researchers have investigated the parameter optimization in LSMs. With the
Chameleon Surface Model (CHASM), Xia et al. (2002, 2004a, 2004b) studied the
adaptability of different parameter optimization methods in LSMs. Their results show
that the parameter optimization in LSMs could improve LSMs in some respects.
Bastidas et al. (2006) adopted the multicriteria method to optimize the parameters in
four different LSMs at five stations. Their researches indicate that the parameter
optimization can improve LSMs. Li et al. (2011a, 2011b) optimized parameters in a
LSM with the expanded CNOP method and their work show that the expanded CNOP
method can improve the simulation ability of the LSM. So, with a suitable method,
the parameter optimization of LSMs could be carried out effectively. There are several
parameter optimization methods suitable to LSMs like the Shuffled Complex
Evolution (SCE-UA) method, the Multistep Auromatic Calibration Scheme (MACS),
the Multi-Objective Complex Evolution (MOCOM-UA) method and the difference
evolution (DE) method. All the prior three methods use a complex method named the
complex evolution method, need massive computation, require tedious programming
and have poor transferability. DE method has a rapid computability, a strong
transferability and a simple designed structure. So, DE method is suitable to execute
the parameter optimization of LSMs.



Soil moisture is important to many hydrological, biogeochemical and biological processes. It plays a key role in processes of the complex soil water heat transmission and vegetation physiological and biochemical, and also has a direct impact on the soil properties, field climate and the decomposition of nutrients. In addition, soil moisture is one of the important conditions for the movement of the microbes in the soil and the breeding of crops. It is affected seriously by the soil property, the atmosphere, the vegetation and so on. If the soil moisture is too high, it is easy to worsen the soil aeration, affect the life action of crops such as the growth and the respiration of the root of crops and the activity of the microbes in the soil. Thereafter, soil moisture can affect plowing and sowing a field, as well as the soil temperature. For government agencies and private companies, who are concerned with weather and climate, geotechnical engineering, flood control, water quality, soil erosion, slope failure, and so on, information about soil moisture is very essential. So the investigation of the ability of land surface models to simulate shallow soil moisture is very important in the aspect of the improvement of environment, the development of the agriculture and the control of desertification.

The conditional nonlinear optimal perturbation (CNOP) method is put forward and developed by Mu et al. (2003, 2010) respectively. The approach of conditional nonlinear optimal perturbation related to parameter (CNOP-P) is used to attain the optimal parameter perturbation, whose nonlinear evolution in the forecast stage gains the optimal value and it is one special case of CNOP. CNOP method has been employed to investigate the response of a grassland ecosystem to climate change and human activities (Mu and Wang, 2007; Sun and Mu, 2011), ENSO predictability (Mu and Duan, 2003; Duan et al., 2004; Duan and Mu, 2006, 2009; Duan et al., 2008; Duan et al., 2009a, 2009b; Duan and Luo, 2010; Duan and Zhang, 2010; Mu et al., 2010), the sensitivity analysis to the eutrophication of lakes (Wang et al., 2012) and so on, and a series of research results are achieved. Sun and Mu (2013, 2014) used the CNOP-P method to study the maximal variation in estimating the net primary production (NPP). The results show that CNOP-P method can capture the characteristic of nonlinear dynamical system As CNOP-P could make the dynamic

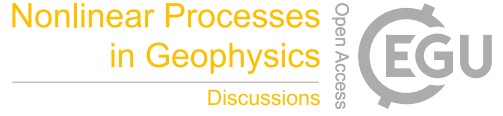

system have the largest nonlinear development in the forecast stage, and it regards
both the sensitivity of the objective parameter to the dynamic system and the
influence of the nonlinearity of the dynamic system to the evolution from the initial
moment to the forecast stage, CNOP-P method could possibly be extended and then
applied to the area of parameter optimization in LSMs. Wang and Huo (2013) used
the extended CNOP-P method and DE method to optimal some parameters in CoLM,
and the results showed that after optimization, the simulation ability of CoLM can be
improved to some degree in the North China Plain. So the expanded CNOP-P method
and DE method are suitable to be employed to optimize the parameters in CoLM. So,
it is reasonable to optimize the parameters with the expanded CNOP-P method and
DE method to improve the simulating ability of soil moisture in the CoLM in the
Hulunbiur Steppe.
As a state-of-the-art land surface model, Common Land Model (CoLM) is
developed by Dai et al. (2001). This model combines the best features of three other
land surface models, such as the Land Surface Model of Bonan (1996), the
Biosphere-Atmosphere Transfer Scheme (BATS) of Dickinson et al. (1993) and the
1994 version of theChinese Academy of Sciences Institute of Atmo-spheric Physics
LSM (IAP94) (Dai and Zeng, 1997). Luo et al. (2008), Xin et al. (2006) , Song et al.
(2009a, 2009b), Zheng et al. (2009), Meng and Cui (2007) simulate China areas with
CoLM, their results indicate that CoLM could simulate China well. So, in this paper,
CoLM is employed to investigate the simulation of soil moisture in the Hulunbiur
Steppe.
The percentage of sand and the percentage of clay in soil are important to the soil
structure and they are different in different kinds of soil. Soil texture, which is related
to the crop production and the field management, is an important soil characteristic
and it is determined by the content of sand, clay and silt in the soil. Soil moisture
could infect the proportion that water drains through the soil, water holding capacity,
soil tilth, organic matter content, and drainage largely. As clayey soil has a larger
water holding capacity than sandy soil, water could move more freely through the
sandy soil than the clayey soil. Furthermore, in CoLM, the thermal conductivity of

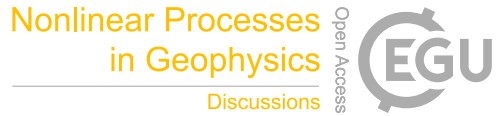
soil solid, the saturated matrix potential, the specific heat capacity of soil solid, the
saturated hydraulic conductivity, and the porosity are produced according to the
percentage of sand and the percentage of clay in soil. Therefore, we chose the
percentage of sand and the percentage of clay as the optimal parameters in CoLM in
this paper.
Actually, we have adopted the expanded CNOP-P method to study the impact of
sand and clay in North China Plain (Wang and Huo, 2013). The results show that the
expanded CNOP-P method and DE method are efficient to optimize the parameters
sand and clay, and the parameters after optimization could make CoLM simulates the
shallow soil moisture better in this area. In addition, we get the conclusion that the
optimization results are affected by the atmospheric forcing data and the observations
of the shallow soil moisture, and the more accurate the data are, the more significant
the optimization results may be. In the purpose of investigating the impact of the
percentage of sand and clay in the soil on the soil moisture in CoLM in Hulunbiur
Steppe, a special arid and semiarid area, and comparing with the results of our former
research, the same three experiments (I-III) as Wang and Huo (2013) are designed. In
this paper, we will check this suspect with these three experiments.
**2. The extension of CNOP-P**
**2.1 Definition of CNOP-P**
In works of Mu et al. (2003, 2010), CNOP method is put forward and developed.
CNOP-P is aimed at the parameter perturbation and it is one special case of CNOP.
Here, we introduce this approach for reader's convenience. Let a nonlinear dynamic
system be described as following equations:
$$\begin{cases} \dfrac{\partial w}{\partial t} = M(w,P), x \in \Omega, t \in [0,T] \\ w|_{t=0} = w_0 \end{cases} \tag{1}$$

where $w(x,t) = (w_1(x,t), w_2(x,t), \cdots, w_m(x,t))$ is a model-state vector, $M$ is a nonlinear
partial differential operator, $\Omega \in \mathbb{R}^n$, $x = (x_1, x_2, \cdots, x_n)$, $0 < T < \infty$, $w_0$ is the initial value,
$P = (P_1, P_2, \cdots, P_l)$ is a parameter vector and $\forall i = 1, 2, \cdots, l$, $P_i$ is a model parameter
invariant with time $t$.



Suppose that $w(\tau)$ is the solution of Eq. (1) at time $\tau$ and $M_\tau$ is the nonlinear
evolution operator from time 0 to time $\tau$ corresponding to the operator $M$, then the
following equation will be established:
$w(\tau) = M_\tau(w_0, P)$.                                    (2)
If $W(T;P)$ is the solution of Eq. (1) at time $T$ corresponding to the parameter
vector $P$ and the initial value $w_0$, and $W(T;P) + w(T;p)$ is the solution of Eq. (1) at
time $T$ corresponding to the initial value $w_0$ and the parameter vector $P+p$, then
the following relations will be established:
$W(T;P) = M_T(w_0, P)$ ,                                    (3)
$W(T;P) + w(T;p) = M_T(w_0, P+p)$ ,                          (4)
where $P$ is a parameter perturbation vector and $w(T;p)$ could indicate the departure
level of the solution of Eq. (1) at time $T$ with the basic state $W(T;P)$, which is
caused by the parameter perturbation vector $p$.
We choose an appropriate norm $\|\cdot\|$ based on the detail physical background. The
objective function under the given constraint condition $\|p\| \leq \delta (\delta > 0)$ is defined as
the following relation:
$J(p) = G(w(T;p))$,                                          (5)
where function $G(\cdot)$ evaluates the departure level which has been described in the
previous paragraph. The parameter perturbation vector $p'$ is the conditional
nonlinear optimal parameter perturbation (CNOP-P), if and only if the parameter
perturbation vector $p'$ satisfies the following relation:
$J(p') = \max\limits_{\|p\| \leq \delta} J(p)$.                              (6)
So CNOP-P is the parameter perturbation vector which could make the objective
function under the given constraint condition attain the maximum, i.e. it is the
parameter perturbation vector that could cause the largest departure level of the
dynamic system at time $T$.
**2.2 Extension of CNOP-P**



In this paper, we use the extension of CNOP-P method proposed by Li et al. (2011a, b)
to optimize the parameter of CoLM. Detailed introduction about the extension of
CNOP-P can also be found in the work of Wang and Huo (2013). Here, we also give a
simple introduction about it.
The parameter perturbation vector $p'$ is considered as the extended CNOP-P if
and only if:
$$J(p') = \max_{P+p \in \omega} J(p) ,$$  (7)
where $P$ means the original parameter vector, $p$ is the parameter perturbation
vector, $\omega$ refers to the value range of the parameter vector and $J(p)$ means the
objective function about $p$ with the following form:
$$J(p) = -G(M_{0 \to T}(w_0, P+p) - O) ,$$  (8)
where $0 \to T$ refers to the time period from time 0 to time $T$ , $M_{0 \to T}(w_0, P+p)$ is the
simulations from time 0 to time $T$ , $O$ is the model state observation vector from
time 0 to time $T$ and the function $G(\cdot)$ evaluates the departure degree between the
simulations and the observations from time 0 to time $T$ .
Therefore, the extended CNOP-P is the parameter perturbation vector satisfying the
constraint condition that could make the simulations closest to the observations. The
method to get the extended CNOP-P is called the extension of CNOP-P method and it
could be employed to conduct the parameter optimization of the land surface model.
**3. The optimization method adopted to calculate CNOP-P**
CoLM is a complex model, and the calculation of CNOP-P with the nonlinear
optimization method depending on the adjoint method would need a lot of
computational resource. It is against the standard that we should employ the
optimization method with a little calculation cost. So we use the differential evolution
method (DE method) as the optimization method in our experiments.
In 1995, Storn and Price (1995) proposed DE method to solve the Chebyshev
Polynomial fitting Problem firstly. Liu et al. (2007) found that DE method is effective
to solve the complex optimization problems. DE method is a parallel, random and



global direct-search algorithm based on the population evolution with the character of
sharing information in populations and remembering group optimal solutions. As a
novel direct search method, DE method use a greedy genetic algorithm to maintain
the excellent population member based on the real number encoding with excellent
convergence properties. Given the randomly generated initial population, DE method
solves the optimization on the basis of the theory of survival of the fittest, in
accordance with the fitness value of every population. It has been applied and
developed by many scholars to solve different problems (He and Wang, 2008; Yu et
al., 2009). Sun and Mu (2009) have found that DE method is effective to obtain
CNOP. Wang and Huo (2013) have used this method to calculate the extension of
CNOP-P effectively. Their work show that it is effective to employ DE method to
handle nonlinear and non-differentiable cost functions if the gradient of the cost
function is hard to obtain or even not obtained. As the cost function about the
parameters may become non-differentiable with the adjustment of the parameter, it is
applicable to optimize the parameters in LSMs with DE method.
**4. Experimental design and the numerical results**
In this paper, NCEP/DOE AMIP-Ⅱ 6-hourly Reanalysis Gaussian Grid data and
NMC Reanalysis 6-hourly surface fluxes data at NCEP/NCAR Center at the
Hulunbiur Steppe $(48.5705°N, 120°E)$ are used. NMC Reanalysis 6-hourly surface
fluxes data, which is currently kept using near real-time observations, is one product
of NCEP/NCAR Reanalysis Ⅰ, which is the first of its kind of National Oceanic and
Atmospheric Administration (NOAA). NCEP/DOE AMIP-Ⅱ 6-hourly surface fluxes
data is one product of NCEP/DOE Reanalysis Ⅱ, which is the second version of
NCEP/NCAR Reanalysis Ⅰ and starts from the beginning of the major satellite era
with a better version of the model used, more observations added and assimilation
errors corrected. Both NCEP/DOE AMIP-Ⅱ 6-hourly Reanalysis Gaussian Grid data
and NMC Reanalysis 6-hourly surface fluxes data are used to investigate the
simulation ability of CoLM to the shallow soil moisture. For convenience, we will call



NCEP/DOE AMIP-Ⅱ 6-hourly Reanalysis Gaussian Grid data dataset Ⅰ and NMC
Reanalysis 6-hourly surface fluxes data dataset Ⅱ in the following part of this paper.
Dataset Ⅰ are the revise of dataset Ⅱ and they are more accurate than dataset Ⅱ.
We will similarly call the percentage of sand in soil sand and the percentage of clay in
soil clay. For the sake of investigating the impact of sand and clay on the shallow soil
moisture, we designed two experiments(Ⅰ and Ⅱ), and simulated the shallow soil
moisture in the following one month with the optimal parameters, which are gained in
the duration of optimization. We select the root mean square deviation as the objective
function with following form:
$$f_r = \sqrt{\frac{\sum\limits_{i=1}^{n}(s_i - o_i)^2}{n}} \, , \tag{9}$$

where $f_r$ refers to the root mean square deviation, $n$ means the integral time steps,
$s_i$ represents the simulation value and $o_i$ is the observation value at time $i$.

### 4.1 Experiment Ⅰ and the results

#### 4.1.1 Experiment Ⅰ

The forcing data in dataset Ⅰ, which contain the precipitation and the large-scale
precipitation, the specific humidity and the air temperature at 10 m above the ground,
the wind component in northward direction and eastward direction at 10 m above the
ground, the atmospheric pressure, the atmospheric longwave radiation and the
incident solar radiation at surface, are used in CoLM in experiment Ⅰ. Here, we took
the forcing data in dataset Ⅰ as the observation data, and compared the data in dataset
Ⅰ with the simulation data of CoLM at the same time.
Generally speaking, CoLM might simulate the shallow soil moisture better after the
double-parameter optimal experiment about sand and clay simultaneously than the
single-parameter optimal experiment about sand or clay. So, we carried out both the
single-parameter optimal experiment related to sand and clay separately and the
double-parameter optimal experiment related to sand and clay simultaneously in this
experiment. Through this experiment, we could check whether the optimal results are
correct in the optimization slot at first, and verify whether the optimal parameters,





which are obtained in the optimization slot, could make CoLM simulate the shallow
soil moisture better at the stage of prediction, and whether the optimal parameters,
which are attained in the double-parameter optimal experiment at the stage of
optimization, could make the simulation ability of CoLM to the shallow soil moisture
the best in the prediction slot. Notice that sand and clay (units: %) are independent
inputs in CoLM, and both in the single-parameter experiment and the
double-parameter experiment, these parameters should satisfy the constraint
condition:
$0 \le \text{sand+clay} \le 100$, $0 \le \text{sand} \le 100$, $0 \le \text{clay} \le 100$.

### 313 4.1.2 The numerical results of experiment Ⅰ

Through a large number of numerical experiments during many different time ranges,
the numerical results show that the optimal parameters could make CoLM simulate
the shallow soil moisture more accurately. Considering the climate change of Inner
Mongolia (Su et al., 2008; Han et al., 2010), in order to better illustrate this
conclusion, we choose the data in 2005 to carry on the process of spin-up and the
process of spin-up reaches a length of 10 years. We choose the time range from May,
2005 to July, 2005 as the time slot of optimization, and choose August, 2005 as the
time slot of prediction in this experiment.

322        Table 1 and Table 2 show the numerical results of experiment Ⅰ. In Table 1 and

Table 2, sand optimization refers to the optimization to sand only, clay optimization
means the optimization to clay only, and sand-clay optimization refers to the
optimization to sand and clay simultaneously. In order to better display the difference
of the shallow soil moisture simulated by CoLM before and after optimization, we
add the mean deviation as the reference function in Table 2. The mean deviation is as
following:

$$f_m = \frac{\sum_{i=1}^{n} |s_i - o_i|}{n} \quad , \tag{10}$$

where $f_m$ refers to the mean deviation, $n$ represents the integral time steps, $s_i$ is



the simulation value at time $i$ and $o_i$ refers to the observation value at time $i$.
Table 1 displays the values of sand and clay before and after optimization, and
Table 2 shows the objective function value and the reference function value before
and after optimization both in the optimization slot and the prediction slot. In Table 2,
$obj1$ refers to the objective function value at the stage of optimization, $obj2$ refers
to the objective function value at the stage of prediction, $ref1$ refers to the reference
function value at the stage of optimization and $ref2$ refers to the reference function
value at the stage of prediction. It is also the case in all subsequent tables. From Table
2, we know that the objective function value and the reference function value are both
smaller after the sand optimization, the clay optimization and the sand-clay
optimization. In addition, it is clear that the extent of diminution is the largest after the
sand-clay optimization in the optimization slot. So CoLM could simulate the shallow
soil moisture better after each optimization and CoLM could simulate the shallow soil
moisture the best after the sand-clay optimization in the optimization slot. Moreover,
in the prediction slot, the root mean square deviation and the mean deviation are both
smaller after the sand optimization, the clay optimization and the sand-clay
optimization. And the objective function value and the reference function value are
both the smallest after the sand-clay optimization at the stage of prediction. That is to
say, CoLM could predict the shallow soil moisture better at the stage of prediction and
the prediction effect is the best after the sand-clay optimization.
To better illustrate the improvement, Figure 1 and Figure 2 are given to
demonstrate the numerical results of experiment Ⅰ graphically. Figure 1 shows the
simulations of the shallow soil moisture before and after sand optimization, the clay
optimization and the sand-clay optimization in the optimization slot and the prediction
slot. Figure 2 shows the corresponding scatter diagram of Figure 1. Here, $obs$ stands
for the observation, $sim1$ means the simulation before the optimization, $sim2$
means the simulation after the sand optimization, $sim3$ means the simulation after
the clay optimization and $sim4$ means the simulation after the sand-clay
optimization. It is also the case in all subsequent figures.





All of the above tables and figures show that, CoLM could simulate the shallow
soil moisture better with the reasonable collocation of sand and clay. Sand is large and
clay is relatively small before the optimization, so the soil water holding capacity is
poor and the soil infiltration capacity is relatively good before the optimization. Hence
it is easy for water to infiltrate into the deep soil layer and the simulation of the
shallow soil moisture of CoLM before the optimization is relatively small. After sand
optimization, sand becomes smaller, the soil water holding capacity is better, and the
simulation of the shallow soil moisture of CoLM is larger than ever. That is to say,
CoLM simulates the shallow soil moisture better after sand optimization. After the
clay optimization, clay is larger and the penetrability of water is worse. So, the
shallow soil moisture simulated by CoLM is larger, and the ability of simulating the
shallow soil moisture of CoLM become better after the clay optimization. Sand and
clay reach an appropriate combination after the sand-clay optimization, and the soil
water holding capacity is better. Therefore, the simulation of the shallow soil moisture
of CoLM is larger and the ability of simulating the shallow soil moisture of CoLM is
better after the sand-clay optimization.
**4.2 Experiment Ⅱ and the results**
**4.2.1 Experiment Ⅱ**
In experiment Ⅱ, the forcing data and the observation data described in the section
4.1.1 are replaced by the corresponding data in dataset Ⅱ. Also, we will conduct the
single-parameter optimal experiment related to sand and clay separately, and the
double-parameter optimal experiment related to sand and clay simultaneously in the
optimization slot. And then we will compare the simulations of CoLM with the
observation data both in the optimization slot and the prediction slot.
**4.2.2 The numerical results of experiment Ⅱ**
Like the results of experiment Ⅰ, by a large amount of numerical experiments for
many different time ranges, the numerical results show that the optimal parameters
could make CoLM simulate the shallow soil moisture better. In order to illustrate this



conclusion more specifically, we also choose the data in 2005 to conduct the process
of spin-up and the process of spin-up reaches a length of 10 years. We still choose the
time slot from May, 2005 to July, 2005 as the optimization slot and August, 2005 as
the prediction slot in experiment Ⅱ.
Table 3 and Table 4 display the results of experiment Ⅱ. In these tables, sand
optimization, clay optimization and sand-clay optimization mean the same as in table
1 and table 2. We also add the mean deviation as the reference function in Table 4, just
like what we have done in Table 2, so that we can better exhibit the difference
between the shallow soil moisture simulated by CoLM before and after optimization.
From Table 4, we can see that the objective function value and the reference function
value are both smaller after sand optimization, clay optimization and sand-clay
optimization. Moreover, we can see that, in the optimization slot, the diminution is the
largest after sand-clay optimization. This means that CoLM could simulate the
shallow soil moisture better after each optimization and, after sand-clay optimization
in the optimization slot, the ability to simulate the shallow soil moisture of CoLM is
the best. In addition, in the prediction slot, the root mean square deviation and the
mean deviation are both smaller after the sand optimization, the clay optimization and
the sand-clay optimization. And the objective function value and the reference
function value are both the smallest after the clay optimization at the stage of
prediction. This means that CoLM could predict the shallow soil moisture better at the
stage of prediction and the prediction effect is the best after the clay optimization.
In order to better illustrate the improvement, we offer Figure 3 and Figure 4 to
display the numerical results of experiment Ⅱ. Figure 3 shows the simulations of the
shallow soil moisture before and after sand optimization, clay optimization and
sand-clay optimization at the optimization slot and the prediction slot. Figure 4 shows
the corresponding scatter diagrams of Figure 3.
Table 3 and Table 4 show that CoLM could simulate the shallow soil moisture
better with the reasonable proportion of sand and clay in soil. Sand is large and clay is
relatively small before the optimization, so the soil water holding capacity is poor and
the soil infiltration is relatively good before the optimization. Therefore water is easy





to infiltrate into the deep soil layer and the simulation of the shallow soil moisture of
CoLM before the optimization is relatively small. After the sand optimization, sand is
smaller and hence the soil water holding capacity is better. Consequently, after the
sand optimization, the simulation of shallow soil moisture of CoLM is larger and
CoLM could simulate the shallow soil moisture better. After the clay optimization,
clay is larger and therefore the penetrating quality of water is worse. For this reason,
the shallow soil moisture simulated by CoLM is larger and the ability of simulating
the shallow soil moisture of CoLM is better after the clay optimization. After the
sand-clay optimization, sand and clay reach an appropriate combination and so the
soil water holding capacity is better. Hence the simulation of the shallow soil moisture
of CoLM is larger and the ability to simulate the shallow soil moisture of CoLM is
better after the sand-clay optimization. All the results of experiments (I and II) are
similar to Wang and Huo (2013), except that, for the NMC Reanalysis 6-hourly
surface fluxes data, at the stage of prediction, the cost function are much smaller than
those in Wang and Huo (2013).
Now let's focus on the peak about the shallow soil moisture in experiments (Ⅰ and
Ⅱ). For this case, we show the convective precipitation and large scale precipitation
in Figure 5. Comparing the precipitation with the shallow soil moisture simulated at
the stage of optimization and the stage of prediction, we find that the precipitation is
almost in accord with the occurrence of the peak, which means the simulations of the
shallow soil moisture are rational.

**4.3 Experiment Ⅲ and the results**

**4.3.1 Experiment Ⅲ**

Latent heat flux, as an important component of Earth's surface energy budget, is the
flux of heat from the Earth's surface to the atmosphere that is associated with
evaporation of water at the surface and subsequent condensation of water vapor in the
troposphere. It is very common to measure latent heat flux with the Bowen ratio
technique, or by eddy covariance.
For testing the ability of CoLM to simulate the latent heat flux, in this experiment,




the shallow soil moisture and the latent heat flux are both chosen as the objective
variables. We choose the weighted root mean square deviation as the objective
function with the following form:
$f_1 = w_1 \cdot f_r(\theta_1) + w_2 \cdot f_r(\theta_2)$ ,                    (11)
where $f_1$ is the weighted root mean square deviation; $\theta_1$ means the shallow soil
moisture, $\theta_2$ is the latent heat flux; $w_1$ and $w_2$, the weight coefficients, satisfy the
condition: $w_1 + w_2 = 1$, and their values are decided based on the dimensions of the
shallow soil moisture and the latent heat flux; $f_r(\theta_1)$ and $f_r(\theta_2)$ represent the root
mean square deviations corresponding to the shallow soil moisture and the latent heat
flux respectively as formulated by Eq. (9).
In this experiment, we use both dataset Ⅰ and dataset Ⅱ to optimize the same
parameters as experiments Ⅰ and Ⅱ in the time slot of optimization. And we predict
the shallow soil moisture and the latent heat flux in the time slot of prediction.

### 4.3.2 The numerical results of experiment Ⅲ

The numerical results of experiment Ⅲ are given in Tables 5-8. Table 5 and Table 6
show the results with dataset Ⅰ, and Table 7 and Table 8 with dataset Ⅱ. We add the
reference function of weighted mean deviation in Table 5 and Table 7 to better display
the variation of the shallow soil moisture before and after optimization. The reference
function of weighted mean deviation refers to the following relation:
$f_2 = w_1 \cdot f_m(\theta_1) + w_2 \cdot f_m(\theta_2)$ ,                    (12)
where $f_2$ means the weighted mean deviation; $\theta_1$ is the shallow soil moisture; $\theta_2$
is the latent heat flux; the weight coefficients, $w_1$ and $w_2$, satisfy the condition:
$w_1 + w_2 = 1$, and their values are chosen based on the dimensions of the shallow soil
moisture and the latent heat flux; $f_m(\theta_1)$ represents the mean deviation corresponding
to the shallow soil moisture formulated by Eq. (10); $f_m(\theta_2)$ means the mean
deviation corresponding to the latent heat flux formulated by Eq. (10).
Table 5 and Table 7 show the objective function values and reference function
values before and after parameter optimization. Table 6 and Table 8 show the
percentage of sand and the percentage of clay in soil before and after parameter



optimization. The results are consistent with the results in experiment Ⅰ and
experiment Ⅱ.
From what we have discussed above, the extended application of CNOP-P method
in CoLM is reasonable and efficient.
**5. Discussions and Conclusions**
From the above three experiments, we can see that the optimized parameters after
sand optimization, clay optimization and sand-clay optimization could make CoLM
simulate the objective variable(s) better at the stage of optimization, and it is the best
after sand-clay optimization. Moreover, the optimized parameters could enable CoLM
to improve its ability of simulation observably at the stage of prediction in these three
experiments. The difference of the numerical results between experiment Ⅰ and
experiment Ⅱ is that, at the stage of prediction, the optimal parameters after the
sand-clay optimization attained in experiment Ⅰ could make CoLM simulate the
shallow soil moisture the best, which is rational, but in experiment Ⅱ, the optimal
parameters attained by clay optimization could make the ability of simulating the
shallow soil moisture of CoLM the best, which is not line with the common sense.
These conclusions are similar to Wang and Huo (2013), although the climate of the
research areas is different, one is a arid and semiarid area, and the other is subhumid
temperate climate. At the same time, dataset Ⅰ are the revise of dataset Ⅱ, and
dataset Ⅰ are more accurate than dataset Ⅱ. As we all know, the simulating ability
of CoLM depends on the accuracy of data, such as forcing data, initial data, boundary
data and so on, and in general, the more accurate the data are, the more reliable the
simulations will be. In our works, we got similar results for both arid and semiarid and
subhumid area, so, we can guess that the factor affecting optimal results and
simulating ability mainly is the accuracy of dataset. The more accurate the dataset are,
the more trustworthy the optimal results will be.
The conclusions obtained in this work also show that the extended application of
CNOP-P method in CoLM is reasonable and efficient. But we just investigated two
special parameters and in one special area, i.e. the Hulunbiur Steppe. It will be





beneficial and helping to lay solid foundations for ecosystem management to consider
more important parameters in CoLM in arid and semiarid areas.



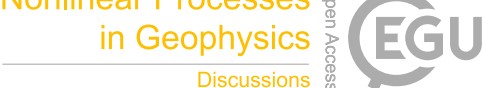

**Author contribution**
Bo Wang designed the experiments and wrote up the manuscript with contributions
from all co-authors, Zhenhua Huo carried the experiments out, and Yujing Yuan and
Shang Wu helped to check the model code and the simulations.



**Acknowledgements**
This research was supported by the National Natural Science Foundation of China
(Grant No. 40805020); the Open Subject of LASG, IAP, CAS(No. 201401); the
foundation for Young University Key Teacher by the Educational Department of
Henan Province (No. 2014GGJS-021); the Emerging Cross and Characteristic
Discipline Plan of Henan University(No. 0000A40450). The authors are grateful to
NOAA/OAR/ESRL PSD, Boulder, Colorado, USA, for the availability of the NCEP
Reanalysis and NCEP_Reanalysis 2 data from their website
(http://www.esrl.noaa.gov/psd).





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



**Figure captions**

FIG.1 Sequence diagram of the shallow soil moisture (unit: $m^3 m^{-3}$) in the experiment

Ⅰ. (a): at the stage of optimization; (b): at the stage of prediction.

FIG.2 Scatter diagram of the shallow soil moisture (unit: $m^3 m^{-3}$) in the experiment Ⅰ.

(a): at the stage of optimization; (b): at the stage of prediction.

FIG.3 Sequence diagram of the shallow soil moisture (unit: $m^3 m^{-3}$) in the experiment

Ⅱ. (a): at the stage of optimization; (b): at the stage of prediction.

FIG.4 Scatter diagram of the shallow soil moisture (unit: $m^3 m^{-3}$) in the experiment Ⅱ.

(a): at the stage of optimization; (b): at the stage of prediction.

FIG.5 Convective precipitation and large scale precipitation in experiment Ⅰ and

experiment Ⅱ. (a): at the stage of optimization in the experiment Ⅰ; (b): at the stage

of prediction in the experiment Ⅰ; (c): at the stage of optimization in the experiment

Ⅱ; (d): at the stage of prediction in the experiment Ⅱ.






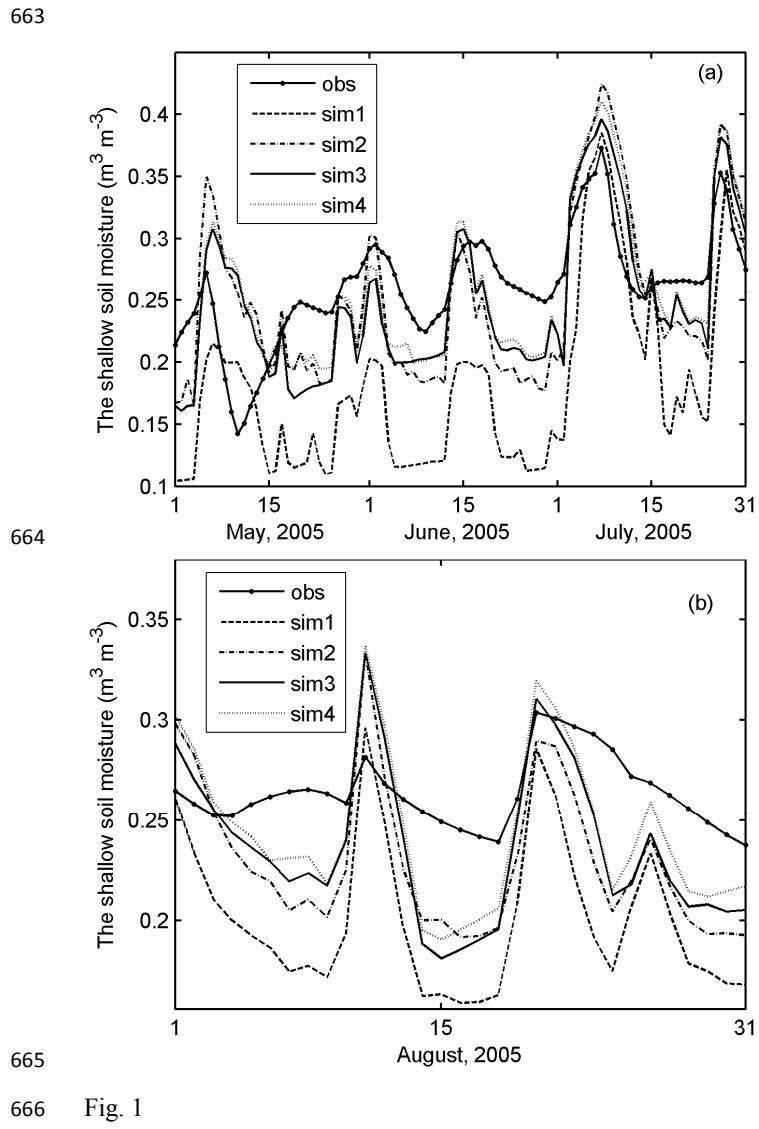



Fig. 1





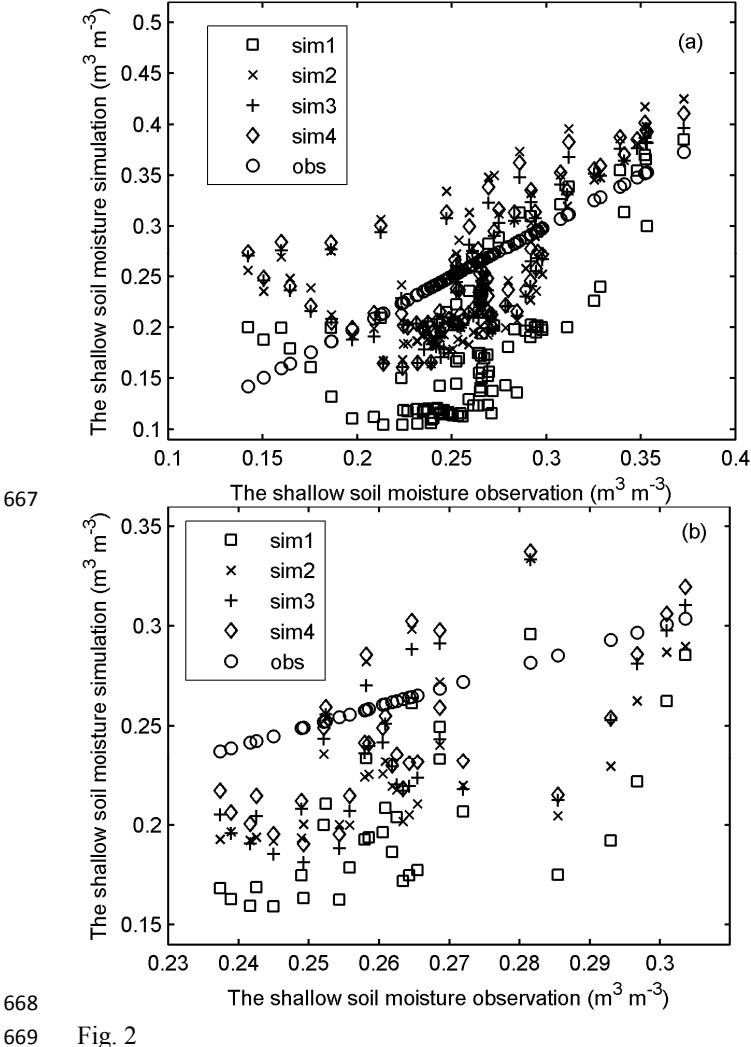



Fig. 2





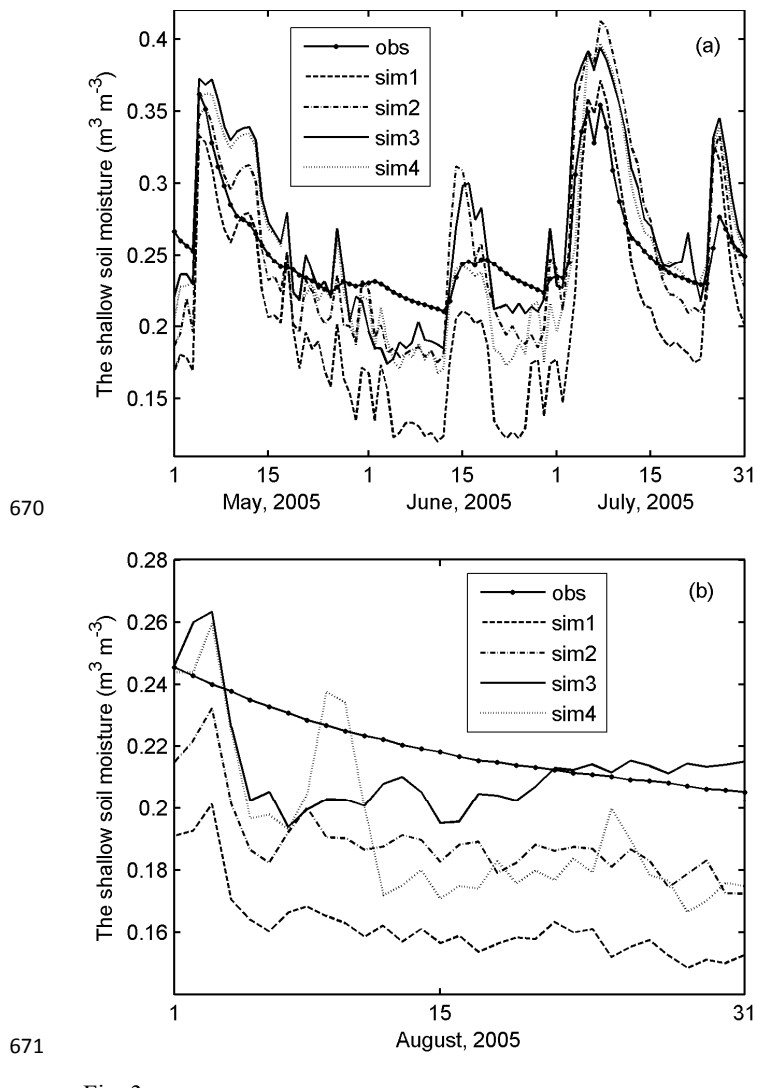



Fig. 3





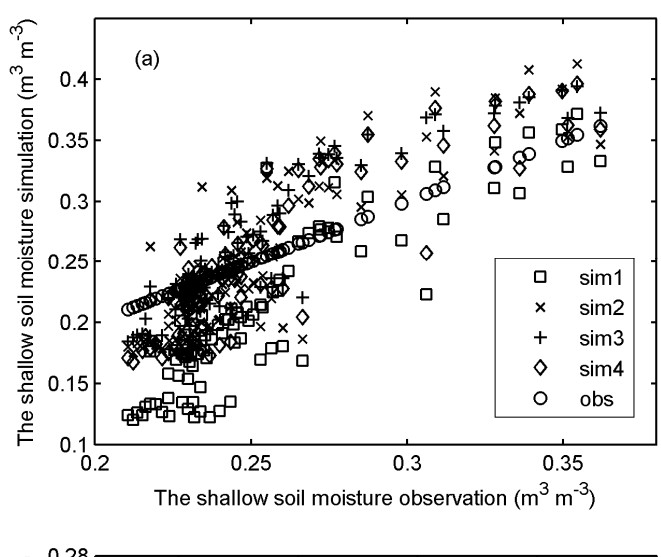


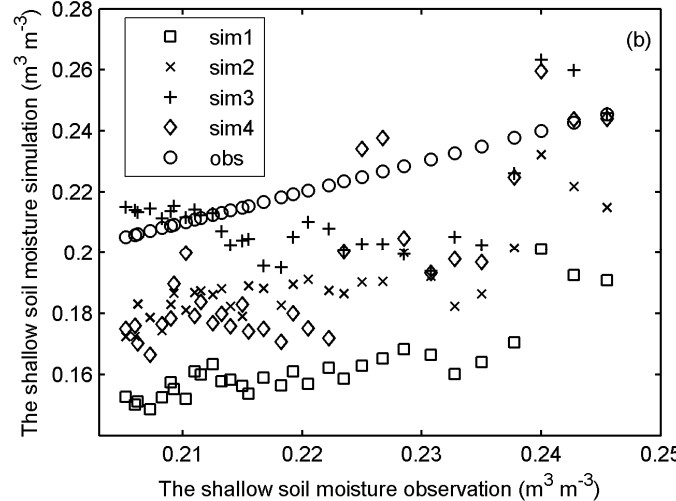


Fig. 4


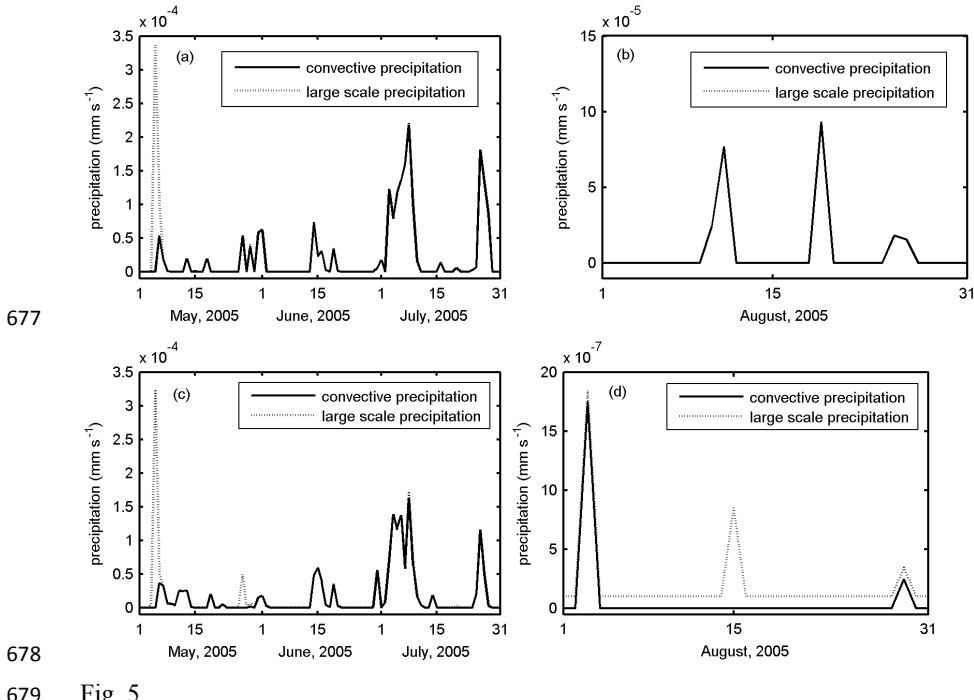



Fig. 5





**Table captions**

Table 1. Comparison of sand and clay (unit: %) before and after optimization in the experiment Ⅰ.

Table 2. Comparison of the objective function values and the referential function values before and after optimization in the experiment Ⅰ.

Table 3. Comparison of sand and clay (unit: %) before and after optimization in the experiment Ⅱ.

Table 4. Comparison of the objective function values and the referential function values before and after optimization in the experiment Ⅱ.

Table 5. Comparison of the objective function values and the referential function values before and after optimization with dataset Ⅰ in the experiment Ⅲ.

Table 6. Comparison of sand and clay (unit: %) before and after optimization with dataset Ⅰ in the experiment Ⅲ.

Table 7. Comparison of the objective function values and the referential function values before and after optimization with dataset Ⅱ in the experiment Ⅱ.

Table 8. Comparison of sand and clay (unit: %) before and after optimization with dataset Ⅱ in the experiment Ⅱ.




Table 1.

| The parameter | Before the optimization | After sand optimization | After clay optimization | After sand-clay optimization |
|---|---|---|---|---|
| Sand | 37.5 | 15.458379 | 37.5 | 28.841991 |
| Clay | 26 | 26 | 43.715421 | 40.281376 |


Table 2.

| Function value | Before the optimization | After sand optimization | After clay optimization | After sand-clay optimization |
|---|---|---|---|---|
| obj1 | 0.098224 | 0.056424 | 0.050492 | 0.049054 |
| ref1 | 0.087392 | 0.050470 | 0.043871 | 0.042262 |
| obj2 | 0.069606 | 0.045452 | 0.040497 | 0.036333 |
| ref2 | 0.063743 | 0.041387 | 0.035004 | 0.031494 |


Table 3.

| The parameter | Before the optimization | After sand optimization | After clay optimization | After sand-clay optimization |
|---|---|---|---|---|
| Sand | 37.5 | 11.215094 | 37.5 | 42.190752 |
| Clay | 26 | 26 | 52.195857 | 54.614659 |


Table 4.

| Function value | Before the optimization | After sand optimization | After clay optimization | After sand-clay optimization |
|---|---|---|---|---|
| obj1 | 0.061491 | 0.038930 | 0.036811 | 0.036528 |
| ref1 | 0.053017 | 0.032882 | 0.030984 | 0.029458 |
| obj2 | 0.058472 | 0.032212 | 0.017816 | 0.033081 |
| ref2 | 0.057880 | 0.030818 | 0.014367 | 0.030414 |





Table 5.

| Function value | Before the optimization | After sand optimization | After clay optimization | After sand-clay optimization |
|---|---|---|---|---|
| obj1 | 0.190404 | 0.135727 | 0.131815 | 0.128480 |
| ref1 | 0.138981 | 0.107459 | 0.105064 | 0.099933 |
| obj2 | 0.135303 | 0.104008 | 0.086822 | 0.083889 |
| ref2 | 0.117531 | 0.083503 | 0.065347 | 0.062849 |


Table 6.

| The parameter | Before the optimization | After sand optimization | After clay optimization | After sand-clay optimization |
|---|---|---|---|---|
| Sand | 37.5 | 11.227582 | 37.5 | 37.489331 |
| Clay | 26 | 26 | 46.279844 | 46.089471 |


Table 7.

| Function value | Before the optimization | After sand optimization | After clay optimization | After sand-clay optimization |
|---|---|---|---|---|
| obj1 | 0.214117 | 0.156524 | 0.152562 | 0.146493 |
| ref1 | 0.169689 | 0.119458 | 0.115074 | 0.110053 |
| obj2 | 0.172814 | 0.150163 | 0.154181 | 0.143004 |
| ref2 | 0.141641 | 0.117755 | 0.122391 | 0.111130 |


Table 8.

| The parameter | Before the optimization | After sand optimization | After clay optimization | After sand-clay optimization |
|---|---|---|---|---|
| Sand | 37.5 | 11.860160 | 37.5 | 25.993371 |
| Clay | 26 | 26 | 36.664604 | 40.290898 |
