# Peer review of "Extended Application of the CNOP-P method in the Inner Mongolia using the"

_Nonlinear Processes in Geophysics, 2016_

## Editor Comment (EC1) · O. Talagrand (Editor) · 30 Jan 2016

I have had a quick look at the paper, and I have noted as Editor several points that will in my opinion need to be dealt with.

1. It is necessary to mention the numerical dimensions of the model CoLM, as well as to give basic information as to what it exactly simulates (state variables, main represented physical processes, …)

2. It is necessary to mention the dimensions of the control vector $p$. It contains sand and clay contents of the soil. But is there only one couple of values, or one couple at each of a number of gridpoints (if yes, how many ?), or what ?

3. The observation set $\{o_i\}$ (eqs 9 and 10) contain quantities which, from what I understand, are expressed in different physical units (see lines 301-305). Are these quantities normalized so as to make them nondimensional and comparable? If yes, how ? If not, the results are unit-dependent, and meaningless.

4. The results of the various minimisations, and especially the minimizing values of the objective and reference functions (tables 2, 4, …) cannot be interpreted if they are not associated with one or several physical measures of the gain obtained through the minimisation. That must be done in terms of quantities such as water content of the soil, precipitation … (but that of course depends on what the model CoLM exactly does).

5. The problem studied by the authors, which they numerically solve with the 'extended CNOP-P' method, is the same problem that is studied in the widely known *variational assimilation*. It is applied here to the identification of 'physical parameters' rather than 'state variables'. And difference with the standard form of variational assimilation is that the minimization is performed through a genetic algorithm (the *DE* method) rather than through a gradient method using the adjoint of the dynamical model. But that is technical, and it must be clearly stated from the start that the problem that is studied is the same as in variational assimilation.

---

## Author Comment (AC1) · 17 Feb 2016

Reply to EC1-supplement:

First of all, all authors of the npg-2016-13 thank Dr. Talagrand for the valuable questions. We'd like to give proper revise to our manuscript in the future. Here, we give the replies as follows:

1. In our research, the Common Land Model is a single-point offline model, which considers the biophysical, biochemical, ecological and hydrological processes. The energy and water transmission among soil, vegetation, snow and atmosphere is well described. This model has one vegetation layer with a realistic photosynthesis-conductance model, 10 unevenly spaced vertical soil layers, and five snow layers (depending on the total snow depth). Model state variables include snow and soil
[Figure]

temperature, ice lens mass and liquid mass in each layer, leaf temperature, canopy water storage, nondimensional snow age, snow-layer number, and snow-layer thickness. This model also has characteristics as follows: Two big leaf model; Two-stream approximation for canopy albedoes calculation; a photosynthesis–stomatal conductance model for sunlit and shaded leaves separately; using a well-built quasi-Newton–Raphson method for simultaneous solution of temperatures of the sunlit and shaded leaves.

2. In our study, the control vector p is a 2-dimension variable, and it contains sand and clay contents of the soil. In the model CoLM, sand and clay (units: %) are independent inputs in CoLM, and for each of the grid point, there is one couple of values for sand and clay contents of the soil. In view of the units, in our experiments, both in the single-parameter experiment and the double-parameter experiment, these parameters should satisfy constraint condition(see line 312)

3. No, we haven't normalized objective and reference functions, and we haven't compared these two functions in our manuscript.

In our research, these two functions are obtained by the same variables, the simulation and observation of shallow soil moisture, the units of these variables are same, and the reference function is also a objective function. For distinguished from the objective function we defined with Eq. (9), we called the function defined with Eq. (10) as reference function. At the same time, the results that we want to verify is, which kind of experiments can get the more accurate optimal results, so, we just compare the three experimental results for objective function and reference function separately. Our results showed that, the conclusions are same for both of these two functions.

4. From the reply to '3.', we can get a guess that, if we choose other kind of objective function, such as associated with physical measures, maybe we can get the same conclusions with the three experiments in our manuscript.

5. Yes, in view of the results of our manuscript, with the 'extended CNOP-P' method,

we can get the minimization of the objective function as variational assimilation method can do. But we also can use this method to obtain the maximal growth of the model which has no adjoint model.

Please also note the supplement to this comment:
http://www.nonlin-processes-geophys-discuss.net/npg-2016-13/npg-2016-13-AC1-supplement.pdf

---

## Referee Comment (RC1) · Anonymous Referee #1 · 9 Mar 2016

The paper describes the applications of an optimisation technique to a land surface model. The method is named conditional nonlinear optimal perturbation about parameters. The land surface model is the Community Land Model. While the paper has merits of applying the CNOP/P method to parameter optimisation that is very useful in complex models, there is a fundamental issue in the choice of parameter being the soil properties. These quantities are part of local physiography and they shall be observed or estimated, but not tuned as effective parameters. If sand and clayare chosen among the tuning parameters it could well be that they will end up compensating other errors in the model or in its uncertain parameters or again in the meteorological forcing. The conclusions on the importance of forcing is obvious and does not represent new insight in the field. I recommend more reflections on the choice of parameters to optimise among non/observable model parameters. The soil moisture results are not very

convincing and I would recommend testing the method first with synthetic data in order to check the result of the optimisation in a case where the truth is known. The use of different forcing in order to argument on the importance of reliable meteorological input is quite essential because the soil could be tuned to compensate the lack of reliable precipitation forcing, and such compensatory tuning may hinder further progress.

Details: P4L44: Global worming...that is interesting and was worth checking wikipendia, which has an entry for "worming" ="moving with difficulty by crawling or wriggling"...I guess is Global warming that was meant here.

---

## Referee Comment (RC2) · Anonymous Referee #2 · 16 Mar 2016

Comments:

I have to point out that this paper (referred as NPGD) is very similar to this one:

Wang, B., and Z. H. Huo, 2013: Extended application of the conditional nonlinear optimal parameter perturbation method in the Common Land Model. Adv. Atmos. Sci., 30(4), 1213–1223, doi: 10.1007/s00376-012-2025-8.

Which is referred as AAS in the following comments. AAS (Advances in Atmospheric Sciences) is a formal IAP journal with IF=1.479.

The editor has mentioned that 22% of the text of the NPGD paper is borrowed from the AAS paper. They have similar title, abstract, keywords, using the same CNOP-P method to optimize the parameters of CoLM. The structures of the two papers are

similar, but the NPGD paper has been paraphrased so that only 22% of text is identified as 'copied'. At the first glance, the only difference between NPGD paper and AAS paper is their study area. After a thoroughly comparison between them, I can't find any significant novel contributions from the NPGD paper.

For my all due respect, I can't accept a paper which is so similar with another peer reviewed paper, even if that one was also written by the same authors. It's impossible to do new works without the foundation of previous researches (by others, or by the authors themselves), but if the authors decide to publish their work, the structure of the manuscript has to be carefully organized to avoid duplication. A well-written paper should be an interesting story, or continuation of another story if you have published another paper talking about the same topic. In my opinion, it is better to organize this paper as follows:

1) In the introduction part, review your previous research and highlight its advantages and disadvantages. Usually one of the disadvantages will by the novel contribution of this paper. You are telling a totally different story, so the literature review should be reorganized to highlight your new contributions in the new paper.

2) If you are using exactly the same method or model, it's unnecessary to repeat their details. It's better to briefly introduce the method with one or two paragraphs, and move the methodology details to appendix, or give some references.

3) The results and discussion section is usually the most important part. If you have done similar work in other area, and you have find something new in another place, it's better to plot the previous results together with the new results in order to intuitively show the differences and similarities between them. Emphasis your research significance comparing with previous researches by others, and by yourselves.

4) In the conclusion part, summary the whole paper and highlight your novel contribution comparing with others works and your previous works. Put a concise 'take home message' at the end of the paper.

[Figure]

Although this paper is similar with the AAS one, the AAS paper has been cited for 8 times (line 139, 171, 180, 222, 257, 430, 432, 492). The authors tried to make some discussion about the novel contributions comparing with the ASS paper, but I don't think they are really novel. It has long been the common sense that the uncertainties of land surface modeling comes from (1) initial/boundary conditions, observation error; (2) parameterization; (3) model structure [Gupta et al., 2005; Duan et al., 2006; Kavetski et al., 2006a, 2006b; Ajami et al., 2007]. In this paper, only 2 parameters in one model were tuned and nothing have been done to the forcing data. There is no evidence to support the conclusion that the input error of forcing data is the main source of uncertainty.

Ajami, N. K., Q. Y. Duan, and S. Sorooshian (2007), An integrated hydrologic Bayesian multimodel combination framework: Confronting input, parameter, and model structural uncertainty in hydrologic prediction, Water Resour. Res., 43, W01403, doi:10.1029/2005wr004745.

Duan, Q. et al. (2006), Model Parameter Estimation Experiment (MOPEX): An overview of science strategy and major results from the second and third workshops, J. Hydrol., 320(1-2SI), 3–17, doi:10.1016/j.jhydrol.2005.07.031.

Gupta, H. V., T. Wagener, and K. J. Beven (2005), Model Calibration and Uncertainty Estimation, in Encyclopedia of hydrological sciences, edited by M. G. Anderson and J. J. McDonnell, pp. 1–17, John Wiley & Sons, Ltd.

Kavetski, D., G. Kuczera, and S. W. Franks (2006a), Bayesian analysis of input uncertainty in hydrological modeling: 1. Theory, Water Resour. Res., 42, W034073, doi:10.1029/2005wr004368.

Kavetski, D., G. Kuczera, and S. W. Franks (2006b), Bayesian analysis of input uncertainty in hydrological modeling: 2. Application, Water Resour. Res., 42, W034083, doi:10.1029/2005wr004376.

Please also note the supplement to this comment:
http://www.nonlin-processes-geophys-discuss.net/npg-2016-13/npg-2016-13-RC2-supplement.pdf
[Figure]

**Supplement:**

ADVANCES IN ATMOSPHERIC SCIENCES, VOL. 30, NO. 4, 2013, 1213–1223

**Extended Application of the Conditional Nonlinear Optimal Parameter Perturbation Method in the Common Land Model**

WANG Bo[*1,2] (王 波) and HUO Zhenhua[2] (霍振华)

[1]*Institute of Applied Mathematics, Henan University, Kaifeng, Henan 475004*

[2]*College of Mathematics and Information Science, Henan University, Kaifeng, Henan 475004*

(Received 10 February 2012; revised 21 October 2012; accepted 23 October 2012)

**ABSTRACT**

An extension of the conditional nonlinear optimal parameter perturbation (CNOP-P) method is applied to the parameter optimization of the Common Land Model (CoLM) for the North China Plain with the differential evolution (DE) method. Using National Meteorological Center (NMC) Reanalysis 6-hourly surface flux data and National Center for Environmental Prediction/Department of Energy (NCEP/DOE) Atmospheric Model Intercomparison Project II (AMIP-II) 6-hourly Reanalysis Gaussian Grid data, two experiments (I and II) were designed to investigate the impact of the percentages of sand and clay in the shallow soil in CoLM on its ability to simulate shallow soil moisture. A third experiment (III) was designed to study the shallow soil moisture and latent heat flux simultaneously. In all the three experiments, after the optimization stage, the percentages of sand and clay of the shallow soil were used to predict the shallow soil moisture in the following month. The results show that the optimal parameters can enable CoLM to better simulate shallow soil moisture, with the simulation results of CoLM after the double-parameter optimal experiment being better than the single-parameter optimal experiment in the optimization slot. Furthermore, the optimal parameters were able to significantly improve the prediction results of CoLM at the prediction stage. In addition, whether or not the atmospheric forcing and observational data are accurate can seriously affect the results of optimization, and the more accurate the data are, the more significant the results of optimization may be.

**Key words**: CNOP-P, parameter optimization, CoLM, shallow soil moisture

**Citation**: Wang, B., and Z. H. Huo, 2013: Extended application of the conditional nonlinear optimal parameter perturbation method in the Common Land Model. *Adv. Atmos. Sci.*, **30**(4), 1213–1223, doi: 10.1007/s00376-012-2025-8.

**1. Introduction**

The conditional nonlinear optimal perturbation (CNOP) method was proposed and subsequently developed by Mu et al (2003) and Mu et al. (2010), respectively. It has been adopted to study ENSO predictability (Mu and Duan, 2003; Duan et al., 2004; Duan and Mu, 2006; Duan et al., 2008; Duan and Mu, 2009; Duan et al., 2009a, b; Duan and Luo, 2010; Duan and Zhang, 2010; Mu et al., 2010), the response of a grassland ecosystem to climate change (Mu and Wang, 2007; Wang et al., 2010; Sun and Mu, 2011; Wang et al., 2011a), and lake eutrophication (Wang et al., 2011b). CNOP-P is a special case of CNOP and is used to obtain the optimal parameter perturbation whose nonlinear evolution attains the optimal value in the forecast moment. Because CNOP-P makes the dynamic system own the largest nonlinear development in the forecast moment, and it considers both the impact of the nonlinearity to the evolution of the dynamic system from the initial moment to the forecast moment and the sensitivity level of the objective parameter to the dynamic system, it is possible to extend the CNOP-P method to apply to the area of parameter optimization in land surface models (LSMs), in order to improve the simulations of the model and study
* * *
*Corresponding author: WANG Bo, wangbo_sdu@163.com

the degree of dependence of some parameter(s) on the model's ability to simulate certain variables. Li et al. (2011a, b) adopted the expanded CNOP method to optimize parameters in a LSM and their results showed that the expanded CNOP method can be successfully used in the parameter optimization of such a model. Thus, it is reasonable to use the expanded CNOP-P method to optimize the parameters in the Common Land Model (CoLM).

The reason that land surface processes capture the attention of scientists and national governments is that they are closely related to human survival and the environment. Land surface processes are a group that includes many of the fundamental physical and biochemical processes which affect atmospheric circulation and climatic change. Those involved researching in this field mainly study the processes closely related to atmospheric movement in the uppermost surface layer, such as complex soil water heat transmission and vegetation physiological and biochemical processes. To better understand land surface processes, it is necessary to improve LSMs. Sun (2005) indicated that parameter values in some physical processes are inaccurate in most LSMs, which could adversely affect their simulations. Therefore, applying parameter optimization to LSMs is very important, as demonstrated by previous work in this area. For example, Bastidas et al. (2006) applied the multicriteria method to four different LSMs at five stations and showed that this method is effective in optimizing the parameters in LSMs. Xia et al. (2002, 2004a, b) studied the applicability of different parameter optimization methods in LSMs using the Chameleon Surface Model (CHASM) and they too demonstrated the benefits of parameter optimization in improving certain aspects of LSMs. Li et al. (2011a, b) adopted the expanded CNOP method to optimize parameters in a LSM, also demonstrating benefits for simulation results. So, we can see that parameter optimization can be highly effective when carried out using a suitable method, and there are several such methods that can be used in the context of LSMs. For example, the Multi-Objective Complex Evolution (MOCOM-UA) method, the Multistep Automatic Calibration Scheme (MACS), the Shuffled Complex Evolution (SCE-UA) method, and the Difference Evolution (DE) method. The first three of these four methods all use the complex evolution method, which requires lengthy programming, huge computational resources, and has poor transferability; while the DE method is computationally much less demanding, it has a simple design structure, and strong transferability. Therefore, the DE method is a relatively good method for conducting parameter optimization in LSMs.

Soil moisture is of fundamental importance to many biological, hydrological and biogeochemical processes. To a variety of government agencies and private companies concerned with runoff potential and flood control, weather and climate, soil erosion and slope failure, geotechnical engineering, reservoir management, and water quality, information about soil moisture is of great value. In terms of controlling the exchange of water and heat energy between the atmosphere and the land surface, soil moisture plays a key role through evaporation and plant transpiration. Therefore, soil moisture is a key variable in the development of the production of precipitation and weather patterns. Using numerical weather prediction models, some studies have shown that improved characterization of surface soil moisture, temperature, and vegetation helps to improve the efficiency of forecast simulations (Li et al., 2011a, b). In addition, soil moisture has a direct impact on the field climate, soil properties and decomposition of nutrients. It is one of the important conditions for the growing of crops and the movement of microbes in the soil, and it is affected greatly by the atmosphere, soil properties, vegetation, and so on. If soil moisture is too high, soil aeration can worsen and adversely affect the activity of microbes in the soil, as well as the health of crops through its influences on processes such as respiration and root growth. Subsequently, aboveground growth will also be affected, leading to lower yields and a greater occurrence of disease. Furthermore, soil moisture can affect abilities to plow and sow a field, as well as the soil temperature, which again is closely linked to crop success. Thus, studying the ability of land models to simulate shallow soil moisture plays an important role in the development of agriculture and afforestation.

The North China Plain is located in eastern China, south of the Great Wall, north of the Huai River, east of the Yellow Sea and the Bohai Sea, and west of the Taihang Mountains and the western hill country. It is the largest plain in China, and most parts of it experience a subhumid temperate climate, with dry conditions in spring and winter, and wet and humid conditions in summer, making it suitable for agricultural production. Despite its climate, the dense human population and high levels of industry in this region result in a great demand for water, meaning drought conditions are a very serious problem, especially in spring, early summer and autumn. Thus, it is particularly important to study soil moisture in the North China Plain.

Developed by Dai et al. (2001), CoLM is a state-of-the-art land surface model that combines the advantages of several other land surface models, such as the Biosphere-Atmosphere Transfer Scheme (BATS)

and NCAR Land Surface Model and it has been successfully coupled to the Community Climate Model (CCM). It has been used to successfully simulate areas of China (Xin et al., 2006; Meng and Cui, 2007; Luo et al., 2008; Song et al., 2009a, b; Zheng et al., 2009), and thus was employed in the present study to simulate the shallow soil moisture in the North China Plain.

Components of soil that determine soil texture, such as the percentages of sand and clay, play very important roles in field management and crop production. Soil texture affects soil properties such as porosity, water holding capacity, organic matter content, and so on. In CoLM (Dai et al., 2001), the specific heat capacity and thermal conductivity of solid soil, its porosity, the saturated matrix potential, and the saturated hydraulic conductivity, are determined by the percentages of sand and clay. Therefore, we selected the percentages of sand and clay as the optimal parameters in CoLM. In order to specifically study the impact of the percentages of sand and clay on soil moisture, we designed both a single- and double-parameter optimal experiment. Using the extended method of CNOP-P, these two experiments were used to study the impacts of the percentages of sand and clay in the North China Plain on the ability of CoLM to simulate shallow soil moisture.

**2. The extension and calculation of CNOP-P**

**2.1 Definition of CNOP-P**

CNOP was proposed by Mu et al (2003), and then further developed by Mu et al. (2010). CNOP-P is a special case of CNOP aimed at parameter perturbation.

Suppose that a nonlinear dynamic system can be described as

$$\begin{cases} \dfrac{\partial \boldsymbol{u}}{\partial t} = N(\boldsymbol{u}, \boldsymbol{P}), \boldsymbol{x} \in \Omega, t \in [0, T] , \\ \boldsymbol{u}|_{t=0} = \boldsymbol{u}_0 , \end{cases} \quad (1)$$

where $\boldsymbol{u}(\boldsymbol{x}, t) = (u_1(\boldsymbol{x}, t), u_2(\boldsymbol{x}, t), \cdots, u_m(\boldsymbol{x}, t))$ is a model-state vector, $\Omega \in R^n$, $N$ is a nonlinear partial differential operator, $\boldsymbol{x} = (x_1, x_2, \cdots, x_n)$, $0 < T < \infty$, $\boldsymbol{u}_0$ is the initial value, $\boldsymbol{P} = (P_1, P_2, \cdots, P_k)$ is a parameter vector and for every $i = 1, 2, \cdots, k, p_i$ is a model parameter invariant with time $t$.

Suppose that $\boldsymbol{M}_\tau$ is the nonlinear evolution operator from time 0 to time $\tau$ corresponding to the operator $\boldsymbol{N}$ and $\boldsymbol{u}(\tau)$ is the solution of Eq. (1) at time $\tau$, then the following equation will be found:

$$\boldsymbol{u}(\tau) = \boldsymbol{M}_\tau(\boldsymbol{u}_0, \boldsymbol{P}) . \quad (2)$$

If $\boldsymbol{U}(T; \boldsymbol{P})$ is the solution of Eqs. (1) at time $T$ corresponding to the initial value $\boldsymbol{u}_0$ and the parameter vector $\boldsymbol{P}$, and $\boldsymbol{U}(T; \boldsymbol{P}) + \boldsymbol{u}(T; \boldsymbol{p})$ is the solution of Eqs. (1) at time $T$ corresponding to the initial value $\boldsymbol{u}_0$ and the parameter vector $\boldsymbol{P} + \boldsymbol{p}$, then the following relations will be found:

$$\begin{cases} \boldsymbol{U}(T; \boldsymbol{P}) = \boldsymbol{M}_T(\boldsymbol{u}_0, \boldsymbol{P}) , \\ \boldsymbol{U}(T; \boldsymbol{P}) + \boldsymbol{u}(T; \boldsymbol{p}) = \boldsymbol{M}_T(\boldsymbol{u}_0, \boldsymbol{P} + \boldsymbol{p}) , \end{cases} \quad (3)$$

where $\boldsymbol{p}$ is a parameter perturbation vector and $\boldsymbol{u}(T; \boldsymbol{p})$ could reflect the degree of departure of the solution of Eqs. (1) at time $T$ with the basic state $\boldsymbol{U}(T; \boldsymbol{p})$, which is caused by the parameter perturbation vector $\boldsymbol{p}$.

In alluding to the detailed physical background, an appropriate norm $||\cdot||$ is selected. The objective function is defined as Eq. (4) under the given constraint condition $||\boldsymbol{p}|| \leqslant \delta \, (\delta > 0)$:

$$\boldsymbol{J}(\boldsymbol{p}) = \boldsymbol{F}(\boldsymbol{u}(T; \boldsymbol{p})) , \quad (4)$$

where function $\boldsymbol{F}(\cdot)$ evaluates the degree of departure, as described above. If and only if the parameter perturbation vector $\boldsymbol{p}'$ satisfies the following relation

$$\boldsymbol{J}(\boldsymbol{p}') = \max_{||\boldsymbol{p}|| \leqslant \delta} \boldsymbol{J}(\boldsymbol{p}) , \quad (5)$$

then the parameter perturbation vector $\boldsymbol{p}'$ can be called the conditional nonlinear optimal parameter perturbation, CNOP-P. So, CNOP-P is the parameter perturbation vector that could make the objective function obtain a maximum under the given constraint condition; that is to say, it is the parameter perturbation vector which could cause the largest degree of departure at time $T$.

**2.2 Extension of CNOP-P**

The CNOP-P method was expanded by Li et al. (2011a, b) and we employ the extension of the CNOP-P method in this paper to conduct the parameter optimization of CoLM.

To obtain the extended CNOP-P method, the detailed expression of the objective function is transformed from Eq. (4):

$$\boldsymbol{J}(\boldsymbol{p}) = \boldsymbol{F}(\boldsymbol{M}_{0 \to T}(\boldsymbol{u}_0, \boldsymbol{P} + \boldsymbol{p}) - \boldsymbol{O}) , \quad (6)$$

where $0 \to T$ refers to the time period from time 0 to time $T$, $\boldsymbol{P}$ is the parameter vector, $\boldsymbol{p}$ is the parameter perturbation vector and $\boldsymbol{O}$ refers to the model state observational data vector from time 0 to time $T$. At the same time, Eq. (5) is changed to be the following relation:

$$\boldsymbol{J}(\boldsymbol{p}') = \min_{\boldsymbol{P} + \boldsymbol{p} \in \omega} J(\boldsymbol{p}) , \quad (7)$$

where $\omega$ refers to the value range of the parameter vector, $\boldsymbol{P}$ is the parameter vector, $\boldsymbol{p}$ is the parameter

perturbation vector and $\boldsymbol{p}'$ refers to the optimal parameter perturbation vector. So, what should be obtained is the optimal parameter perturbation vector satisfying the constraint condition, which could make the objective function attain a minimum during the whole integral procedure. The parameter perturbation vector $\boldsymbol{p}'$ is the extended CNOP-P, and the method to obtain the extended CNOP-P is called the extended CNOP-P method, which could be used to optimize the parameters in a LSM.

**2.3   *The optimization method adopted to calculate CNOP-P**

As CoLM is a very large model, using the nonlinear optimization method based on the adjoint method would require a lot of computational resource. However, the required standards would not be met if we were to adopt the optimization method with a small amount of calculation cost, and so we adopted the DE method as the optimization method in our experiments.

The DE method was proposed by Storn and Price (1995) and was first used to solve the Chebyshev Polynomial Fitting Problem, before later being found to be effective in solving complex optimization problems (Liu et al., 2007). The DE method is a random, parallel and global direct-search algorithm based on population evolution with the feature of remembering group optimal solutions and sharing information in populations. It is a novel direct search method using a greedy criterion to make a decision, and is based on the real number encoding with excellent convergence properties. The DE method solves the optimization problem through the cooperation and competition among the populations. It has been applied and developed by many scholars (e.g. He and Wang, 2008; Yu et al., 2009) and, furthermore, has been found to be effective in computing CNOP (Sun and Mu, 2009). Their results showed that the DE algorithm is effective at handling non-differentiable and nonlinear cost functions when the gradient of the cost function is not obtained or hard to obtain. When the parameters are tuned, the cost function about the parameter may be non-differentiable, so it is reasonable to use the DE method to optimize the parameters in LSMs.

The DE method begins first with an initial population generated randomly, and then carries through the continuous iterative computations according to certain operation rules. Finally, the DE method demands that we should direct the search process to be close to the optimal solution through the method of survival of the fittest, according to the fitness value of every population.

**3.   The numerical experiments**

In this paper, National Meteorological Center (NMC) Reanalysis 6-hourly surface flux data and National Center for Environmental Prediction/Department of Energy (NCEP/DOE) Atmospheric Model Intercomparison Project-II (AMIP-II) 6-hourly Reanalysis Gaussian Grid data from the NCEP/NCAR Center at the North China Plain $(35.2375°N, 118.125°E)$ are used to study the ability of CoLM to simulate shallow soil moisture. The NMC Reanalysis 6-hourly surface data comprise one product of NCEP/NCAR Reanalysis I, which was the first of its kind at NOAA (National Oceanic and Atmospheric Administration), and is a dataset that is kept current using near real-time observational data. NCEP/DOE AMIP-II 6-hourly surface flux data comprise one product of NCEP/DOE Reanalysis II, which is the second version of NCEP/NCAR Reanalysis I, and started at the beginning of the major satellite era, with more observational data added, assimilation errors corrected and a better version of the model used. NCEP/NCAR Reanalysis data have been applied and studied by many scholars (e.g. Xie et al., 2008; Fang et al., 2010; Xie et al., 2010), and results have shown that these data are efficient to use, albeit with some data uncertainties existing.

For convenience, we refer to the NMC Reanalysis 6-hourly surface flux data as dataset I and the NCEP/DOE AMIP-II 6-hourly Reanalysis Gaussian Grid data as dataset II in the following part of the paper, and we will also refer to the percentage of sand in soil simply as 'sand' and the percentage of clay in soil as 'clay' for short. Note that the original values of sand and clay were obtained from the Food and Agriculture Organization (FAO) and State Soil Geographic (STATSGO) datasets, combined. In order to investigate the impact of sand and clay on shallow soil moisture, we designed two experiments (I and II), and then predicted the shallow soil moisture in the following month with the optimal parameters obtained by the numerical experiment. We chose the root mean square deviation as the objective function with the following form:

$$f_{\mathrm{r}} = \sqrt{\frac{\sum\limits_{i=1}^{n}(s_i - o_i)^2}{n}} \, , \qquad (8)$$

where $s_i$ represents the simulation value at time $t = t_i$, $o_i$ refers to the observational data at time $t = t_i$, $n$ represents the integral time steps and $f_{\mathrm{r}}$ refers to the root mean square deviation.

As we know, the latent heat flux is closely related to rainfall and land surface evapotranspiration processes.

As an important component of the water balance of the land surface, land surface evapotranspiration is closely related to soil components, soil environment, vegetation ecological physiological processes etc. Therefore, it is worthwhile studying the impact of sand and clay in soil on the latent heat flux in CoLM.

Soil moisture plays a very critical role in the variation of climate, and is an important variable in the study of land surface processes. It affects climate variation through changing sensible heat flux, latent heat flux and longwave radiation fluxes transferred from the land surface to the atmosphere. Variation of soil moisture could also affect the thermal properties of the soil and hydrological processes, as well as change different parameters of the land surface. Thus, shallow soil moisture and latent heat flux influence each other, and are closely related to soil texture. In order to further investigate the impacts of sand and clay on shallow soil moisture and latent heat flux, we designed a third experiment (experiment III), the detail of which, along with experiments I and II, follows below.

**3.1 Experiment I**

In order to drive CoLM, the wind component in the eastward and northward directions at 10 m above the land surface, air temperature and specific humidity at 10 m above the land surface, incident solar radiation and the atmospheric longwave radiation at the surface, atmospheric pressure, precipitation, and large-scale precipitation in dataset I were used as the forcing data. Furthermore, we took the observational data in dataset I at the same time as the offset information to the simulation data of CoLM.

In general, CoLM might simulate shallow soil moisture better with optimized parameters obtained after the double-parameter optimal experiment on sand and clay simultaneously, rather than the single-parameter optimal experiment either on sand or clay. For studying the above issues, we designed the single-parameter optimal experiment on sand and clay separately, and a double-parameter optimal experiment on sand and clay simultaneously. Through this experiment, we would be able to verify whether the optimal results are correct in the optimization slot, whether the optimized parameter obtained in the optimization slot could improve the ability of CoLM to simulate shallow soil moisture at the stage of prediction, and whether the optimized parameter obtained after the double-parameter experiment in the optimization slot could enable CoLM to simulate shallow soil moisture best at the stage of prediction. Note that sand and clay are independent inputs in CoLM and, both in the single- and double-parameter experiment, sand and clay (units: %) should satisfy the constraint condition:

$0 \leqslant \text{sand+clay} \leqslant 100, 0 \leqslant \text{sand} \leqslant 100, 0 \leqslant \text{clay} \leqslant 100.$

**3.2 Experiment II**

Compared with experiment I, in experiment II the forcing data and the observational data were replaced by the data in dataset II; everything else was the same as in experiment I.

**3.3 Experiment III**

In this experiment, shallow soil moisture and latent heat flux were chosen as the objective variables. We chose the weighted root mean square deviation as the objective function with the following form:

$$f_1 = w_1 f_{\mathrm{r}}(\theta_1) + w_2 f_{\mathrm{r}}(\theta_2) , \qquad (9)$$

where $\theta_1$ is shallow soil moisture; $\theta_2$ is latent heat flux; the weight coefficients, $w_1$ and $w_2$, satisfy the condition: $w_1 + w_2 = 1$, and their values are chosen based on the dimensions of the shallow soil moisture and latent heat flux; $f_{\mathrm{r}}(\theta_1)$ is the root mean square deviation corresponding to shallow soil moisture as formulated by Eq. (8); $f_{\mathrm{r}}(\theta_2)$ is the root mean square deviation corresponding to latent heat flux as formulated by Eq. (8); and $f_1$ is the weighted root mean square deviation.

In this experiment, we used both dataset I and dataset II to optimize the same parameters as experiments I and II in the time slot of optimization. Furthermore, we predicted the shallow soil moisture and latent heat flux in the time slot of prediction.

**4. Results**

The results of experiments I and II show that the optimized parameters are able to make the simulation values of shallow soil moisture attained by CoLM more accurate through a large amount of numerical experiments during many time slots. In order to evidence this conclusion, we chose data for 1992 to conduct the process of spin-up and the time period from February 1992 to April 1992 as the time slot of optimization, with May 1992 as the time slot of prediction. With the forcing data as the model input, the process of spinning up the model reached a length of 10 years. We chose the same time slots for optimization and prediction in experiment III.

**4.1 Experiment I**

The results of experiment I are shown in Tables 1 and 2. To better display the variation of shallow soil moisture before and after optimization, we include the reference function of mean deviation in Table 1. The reference function of mean deviation refers to the following relation:

**Table 1.** Comparison of the objective and reference function values before and after optimization in experiment I.

| Function value | Before optimization | After sand optimization | After clay optimization | After sand-clay optimization |
|---|---|---|---|---|
| obj1 | 0.081264 | 0.040957 | 0.026151 | *0.022021* |
| ref1 | 0.076789 | 0.034808 | 0.019441 | *0.015564* |
| obj2 | 0.085325 | 0.044432 | *0.026024* | 0.040908 |
| ref2 | 0.080919 | 0.039968 | *0.019227* | 0.031388 |

**Table 2.** Comparison of sand and clay (units: %) before and after optimization in experiment I.

| Parameter | Before optimization | After sand optimization | After clay optimization | After sand-clay optimization |
|---|---|---|---|---|
| Sand | 43 | 0.000556 | 43 | 36.096822 |
| Clay | 18 | 18 | 48.695987 | 55.918477 |

$$f_{\mathrm{m}} = \frac{\sum\limits_{i=1}^{n} |s_i - o_i|}{n} , \qquad (10)$$

where $f_{\mathrm{m}}$ refers to the mean deviation.

Table 1 shows that the root mean square deviation value and the mean deviation value of the shallow soil moisture are both smaller after sand optimization, clay optimization and sand-clay optimization. In Table 1, obj1 refers to the objective function value at the stage of optimization, obj2 is the objective function value at the stage of prediction, ref1 stands for the reference function value at the stage of optimization, and ref2 refers to the reference function value at the stage of prediction. This is also the case in all subsequent tables. For ease of reference, the smallest function value of each line in the table is italicized, which is also the case in Tables 3, 5 and 7. From Table 1, it is clear that the extent of diminution is largest after sand-clay optimization in the optimization slot from February 1992 to April 1992, meaning the ability of CoLM to simulate shallow soil moisture is better after these optimizations and is best after sand-clay optimization. In contrast with values before the optimizations, the root mean square deviation and mean deviation are both smaller after sand optimization, clay optimization and sand-clay optimization, and the extent of diminution is largest after clay optimization at the stage of prediction in May 1992. This means the ability of CoLM to predict shallow soil moisture is better at the stage of prediction and it is best after clay optimization. Table 2 gives a comparison of sand and clay (units: %) before and after optimization in experiment I.

In order to further demonstrate the improvement, Figs. 1–4 exhibit graphically the ability of CoLM to simulate shallow soil moisture before and after optimization. Figures 1 and 2 show the simulations of shallow soil moisture before and after optimization in the optimization slot. Figures 3 and 4 show the same

simulations in the prediction slot. In Figs. 1–4, obs

[Figure]

**Fig. 1.** Shallow soil moisture (units: $\mathrm{m^3 m^{-3}}$) during the stage of optimization in experiment I.

[Figure]

**Fig. 2.** Scatter diagram of shallow soil moisture (units: $\mathrm{m^3 m^{-3}}$) during the stage of optimization in experiment I.

[Figure]

**Fig. 3.** Shallow soil moisture (unit: $m^3 m^{-3}$) at the stage of prediction in the experiment I.

[Figure]

**Fig. 4.** Scatter diagram of shallow soil moisture (units: $m^3 m^{-3}$) during the stage of prediction in experiment I.

stands for observation, sim1 refers to simulation before optimization, sim2 to simulation after sand optimization, sim3 to simulation after clay optimization, and sim4 to simulation after sand-clay optimization.

The above tables and figures show that the ability of CoLM to simulate shallow soil moisture is better with reasonable collocation of sand and clay. In addition, clay is too small and sand is relatively large before optimization, so soil infiltration is relatively good and

the soil water holding capacity is relatively poor; it is easy for water to infiltrate into deep soil layers. Hence, the simulation of shallow soil moisture by CoLM is relatively small before optimization. After sand optimization, sand is smaller and the soil water holding capacity is better. Therefore, the simulation of shallow soil moisture by CoLM is larger and CoLM is able to simulate shallow soil moisture better after sand optimization. Clay is larger after clay optimization and so it is hard for water to infiltrate into deep soil layers. Hence, the simulation of shallow soil moisture is larger and the simulation ability of CoLM is better after clay optimization. Sand and clay reaches an appropriate collocation after sand-clay optimization, and then the soil water holding capacity is better. Therefore, the simulation of shallow soil moisture is larger and the simulation ability is better after sand-clay optimization. Consequently, the extended application of CNOP-P in CoLM is reasonable and effective.

The above results show that the simulation ability improves after parameter optimization, and is best after sand-clay optimization at the stage of optimization. In addition, the optimal parameters enable CoLM to predict shallow soil moisture better at the stage of prediction, and the optimal parameters after clay optimization could make CoLM predict shallow soil moisture the best, rather than sand-clay optimization, which is not reasonable.

**4.2 Experiment II**

The results of experiment II are displayed in Tables 3 and 4. Table 3 shows that the root mean square deviation and mean deviation of shallow soil moisture are both smaller in the optimization slot from February 1992 to April 1992 after sand optimization, clay optimization and sand-clay optimization, and they are smallest after sand-clay optimization. This means that the simulation ability is better after optimization and is best after sand-clay optimization in the optimization slot. The same results can also be founded for the stage of prediction in May 1992. Therefore, the optimal parameters enable CoLM to better simulate shallow soil moisture and the simulation ability is best after sand-clay optimization in the stage of prediction. Table 4 gives a comparison of sand and clay (units: %)

**Table 3.** Comparison of the objective and reference function values before and after optimization in experiment II.

| Function value | Before optimization | After sand optimization | After clay optimization | After sand-clay optimization |
|---|---|---|---|---|
| obj1 | 0.094034 | 0.055313 | 0.032480 | *0.031982* |
| ref1 | 0.089464 | 0.050779 | 0.024247 | *0.024080* |
| obj2 | 0.098777 | 0.065846 | 0.042375 | *0.040925* |
| ref2 | 0.089175 | 0.061338 | 0.037062 | *0.036204* |

**Table 4.** Comparison of sand and clay (units: %) before and after optimization in experiment II.

| Parameter | Before optimization | After sand optimization | After clay optimization | After sand-clay optimization |
|---|---|---|---|---|
| Sand | 43 | 0.106261 | 43 | 46.512011 |
| Clay | 18 | 18 | 51.617713 | 53.485161 |

before and after optimization in experiment II.

Again, in order to further demonstrate the improvement graphically, we present Figs. 5–8. Figures 5 and 6 show the simulations of shallow soil moisture before and after optimization in the optimization slot. Figures 7 and 8 show the same simulations in the prediction slot. In Figs. 5–8, obs, sim1, sim2, sim3 and sim4 have the same meaning as in Figs. 1–4.

The above mentioned tables and figures demonstrate that the appropriate collocation of sand and clay is able to improve the simulation ability. In addition, clay is too small and sand is relatively large before optimization, so that soil infiltration is relatively good and the soil water holding capacity is relatively poor; it is easy for water to infiltrate into deep soil layers. Hence, the simulation of shallow soil moisture is relatively small before optimization. Sand is smaller after sand optimization, and so the soil water holding capac-

[Figure]

**Fig. 5.** Shallow soil moisture (units: $m^3\,m^{-3}$) during the stage of optimization in experiment II.

[Figure]

**Fig. 7.** Shallow soil moisture (units: $m^3\,m^{-3}$) during the stage of prediction in experiment II.

[Figure]

**Fig. 6.** Scatter diagram of shallow soil moisture (units: $m^3\,m^{-3}$) during the stage of optimization in experiment II.

[Figure]

**Fig. 8.** Scatter diagram of shallow soil moisture (units: $m^3\,m^{-3}$) during the stage of prediction in experiment II.

ity is better. Therefore, the simulation is larger and the simulation ability is better after sand optimization. Clay is larger after clay optimization, meaning it is hard for water to infiltrate into deep soil layers. Hence, the simulation is larger and simulation ability is better after clay optimization. Sand and clay reaches an appropriate collocation after sand-clay optimization, and then the soil water holding capacity is better. Therefore, the simulation is larger and the simulation ability is better after sand-clay optimization. Consequently, the extended application of CNOP-P in CoLM is reasonable and effective.

The above results show that the simulation ability becomes better after parameter optimization, and is best after sand-clay optimization during the stage of optimization. In addition, the optimal parameters enable CoLM to better predict shallow soil moisture during the stage of prediction, and the optimal parameters after sand-clay optimization lead to the best

prediction ability, which one would expect.

**4.3 *experiment III**

The results of experiment III are shown in Tables 5–8. Tables 5 and 6 display the results with dataset I, and Tables 7 and 8 with dataset II. To better display the variation of shallow soil moisture before and after optimization, the reference function of weighted mean deviation was included in Tables 5 and 7. The reference function of weighted mean deviation refers to the following relation:

$$f_2 = w_1 f_m(\theta_1) + w_2 f_m(\theta_2) , \qquad (11)$$

where $w_1$ and $w_2$, satisfy the condition $w_1 + w_2 = 1$, and their values are chosen based on the dimensions of shallow soil moisture and latent heat flux; $f_m(\theta_1)$ is the mean deviation corresponding to shallow soil moisture formulated by Eq. (10); $f_m(\theta_2)$ is the mean deviation corresponding to latent heat flux formulated

**Table 5.** Comparison of the objective and reference function values before and after optimization with dataset I in experiment III.

| Function value | Before optimization | After sand optimization | After clay optimization | After sand-clay optimization |
|---|---|---|---|---|
| obj1 | 0.131300 | 0.091818 | 0.073891 | *0.067639* |
| ref1 | 0.115039 | 0.074113 | 0.055140 | *0.049222* |
| obj2 | 0.173083 | 0.130610 | *0.113738* | 0.127728 |
| ref2 | 0.150622 | 0.107103 | *0.086285* | 0.098819 |

**Table 6.** Comparison of sand and clay (units: %) before and after optimization with dataset I in experiment III.

| The parameter | Before the optimization | After sand optimization | After clay optimization | After sand-clay optimization |
|---|---|---|---|---|
| Sand | 43 | 0.008010 | 43 | 36.096611 |
| Clay | 18 | 18 | 49.900752 | 55.918330 |

**Table 7.** Comparison of the objective and reference function values before and after optimization with dataset II in experiment III.

| Function value | Before optimization | After sand optimization | After clay optimization | After sand-clay optimization |
|---|---|---|---|---|
| obj1 | 0.148110 | 0.108424 | 0.084713 | *0.084033* |
| ref1 | 0.129293 | 0.089948 | 0.062946 | *0.062072* |
| obj2 | 0.198339 | 0.170553 | 0.143120 | *0.140667* |
| ref2 | 0.160663 | 0.136113 | 0.108906 | *0.106713* |

**Table 8.** Comparison of sand and clay (units: %) before and after optimization with dataset II in experiment III.

| Parameter | Before optimization | After sand optimization | After clay optimization | After sand-clay optimization |
|---|---|---|---|---|
| Sand | 43 | 0.105718 | 43 | 46.277991 |
| Clay | 18 | 18 | 50.837421 | 53.710283 |

by Eq. (10); and $f_2$ is the weighted mean deviation.

Tables 5 and 7 show the objective function and reference function values before and after parameter optimization. Tables 6 and 8 show the sand and clay in soil before and after parameter optimization. The results are consistent with those of experiments I and II.

**5.  Conclusion**

According to the results of all three experiments conducted in this study, we have been able to show that the optimized parameters after sand optimization, clay optimization and sand-clay optimization enable CoLM to better simulate the objective variable(s) during the stage of optimization, and is best after sand-clay optimization. Moreover, the optimized parameters improved the simulation ability notably during the stage of prediction in each of the three experiments. With dataset II, the optimized parameters after sand-clay optimization could be used to best predict the simulations of the objective variable(s) during the stage of prediction. However, with dataset I the optimized parameters after sand-clay optimization improved simulations less than after clay optimization during the stage of prediction, which is an unexpected result. As dataset II is more accurate than dataset I, we conclude that the more accurate the data used by CoLM, the more effective the parameter optimization of CoLM may be. In future work, we will use field observational data as the forcing data and offset information to further study how parameter optimization in CoLM affects its ability to simulate different variables in arid and semiarid areas, thus checking whether the above conclusion holds true.

From the above conclusions, we can see that the extended application of CNOP-P in CoLM is reasonable and effective. However, in this study we have investigated the optimization of two specific parameters, and in one specific area, i.e. the North China Plain. Other important parameters should also be taken into account; plus, applications of this method in arid and semiarid areas of North China would be beneficial, helping to lay solid foundations for ecosystem management.

***Acknowledgements.***  This research was supported by the National Natural Science Foundations of China (Grant Nos.  40805020 and 10901047) and the Natural Science Foundation of Henan Province (Grant No. 112300410054). The authors are grateful to NOAA/OAR/ ESRL PSD, Boulder, Colorado, USA, for the availability of the NCEP Reanalysis and NCEP_Reanalysis 2 data from their website (http://www.esrl.noaa.gov/psd/).

**REFERENCES**

Bastidas, L. A., T. S. Hogue, S. Sorooshian, H. V. Gupta, and W. J. Shuttleworth, 2006: Parameter sensitivity analysis for different complexity land surface models using multicriteria methods. *J. Geophys. Res.*, **111**, doi: 10.1029/2005JD006377.

Dai, Y. J., and Coauthors, 2001: Common land model (CLM): Technical documentation and user's guide. [Available online at http://globalchange.bnu.edu.cn/ research/models.]

Duan, W. S., and M. Mu, 2006: Investigating decadal variability of El Niño–Southern Oscillation asymmetry by conditional non-linear optimal perturbation. *J. Geophys. Res.*, **111**, C07015, doi: 10.1029/2005JC 003458.

Duan, W. S., and M. Mu, 2009: Conditional nonlinear optimal perturbation: Applications to stability, sensitivity, and predictability. *Sci. China* (*D*), **52**, 7, 883–906.

Duan, W. S., and H. Y. Luo, 2010: A new strategy for solving a class of nonlinear optimization problems related to weather and climate predictability. *Adv. Atmos. Sci.,* **27**, 741–749, doi: 10.1007/s00376-009-9141-0.

Duan, W. S., and R. Zhang, 2010: Is model parameter error related to spring predictability barrier for El Niño events? *Adv. Atmos. Sci.*, **5**, 1003–1013.

Duan, W. S., M. Mu, and B. Wang, 2004: Conditional nonlinear optimal perturbations as the optimal precursors for El Niño–Southern Oscillation events. *J. Geophys. Res.*, **109**, D23105, doi: 10.1029/2004JD004756.

Duan, W. S., H. Xu, and M. Mu, 2008: Decisive role of nonlinear temperature advection in El Niño and La Niña amplitude asymmetry. *J. Geophys. Res.*, **113**, C01014. doi: 10.1029/2006JC003974.

Duan, W. S., F. Xue, and M. Mu, 2009a: Investigating a nonlinear characteristic of ENSO events by conditional nonlinear optimal perturbation. *Atmospheric Research*, **94**, 10–18.

Duan, W. S., X. C. Liu, K. Y. Zhu, and M. Mu, 2009b: Exploring the initial error that causes a significant spring predictability barrier for El Niño events. *J. Geophys. Res.*, **114**, C04022. doi: 10.1029/2008JC004925.

Fang, Z. F., J. Lei, X. N. Lu, X. Qu, and Q. Li, 2010: A comparison of 500 hPa field of East-Asia between the NCEP/NCAR reanalysis data and the ERA-40 of ECMWF data. *Acta Meteorologica Sinica*, **68**(2), 270–276. (in Chinese)

He, Y. C., and X. Z. Wang, 2008: Solution of hard constrained optimization problem based on modified differential evolution algorithm. *Computer Engineering*, **34**, 193–217. (in Chinese)

Li, H. Q., W. D. Guo, G. D. Sun, and Y. C. Zhang, 2011a: Using conditional nonlinear optimal perturbation method in parameter optimization of land surface processes model. *Acta Physica Sinica*, **60**,

019201, 1–7. (in Chinese)

Li, H. Q., W. D. Guo, G. D. Sun, Y. C. Zhang, and C. B. Fu, 2011b: A new approach for parameter optimization in land surface model. *Adv. Atmos. Sci.*, **28**(5), 1056–1066, doi: 10.1007/s00376-010-0050-z.

Liu, B., L. Wang, and Y. H. Jin, 2007: Advances in differential evolution. *Control and Decision*, **22**, 721–729. (in Chinese)

Luo, S. Q., S. H. Lü, Y. Zhang, Z. Y. Hu, Y. M. Ma, S. S. Li, and L. Y. Shang, 2008: Simulation analysis on land surface process of BJ site of central Tibetan Plateau using CoLM. *Plateau Meteorology*, **27**, 259–271. (in Chinese)

Meng, C. L., and J. Y. Cui, 2007: Study on soil evaporation and coupling transmission of soil moisture and heat in arid areas. *Arid Zone Research*, **24**, 141–145. (in Chinese)

Mu, M., and W. S. Duan, 2003: A new approach to studying ENSO predictability: Conditional nonlinear optimal perturbation. *Chinese Science Bulletin*, **48**, 1045–1047.

Mu, M., and B. Wang, 2007: Nonlinear instability and sensitivity of a theoretical grassland ecosystem to finite-amplitude perturbations. *Nonlinear Processes Geophysics*, **14**, 409–423.

Mu, M., W. S. Duan, and B. Wang, 2003: Conditional nonlinear optimal perturbation and its applications. *Nonlinear Processes in Geophysics*, **10**, 493–501.

Mu, M., W. S. Duan, Q. Wang, and R. Zhang, 2010: An extension of conditional nonlinear optimal perturbation approach and its applications. *Nonlinear Processes in Geophysics*, **17**, 211–220.

Song, Y. M., W. D. Guo, and Y. C. Zhang, 2009a: Performances of CoLM and NCAR_CLM3.0 in simulating land-atmosphere interactions over typical forest ecosystem in China. Part Ⅰ: Preliminary analysis of the simulations based on diff- erent models. *Climatic and Environmental Research*, **14**, 229–242. (in Chinese)

Song, Y. M., W. D. Guo, and Y. C. Zhang, 2009b: Performances of CoLM and NCAR_CLM3.0 in simulating land-atmosphere interactions over typical forest ecosystem in China. Part Ⅱ: Impact of different parameterization schemes on simulations. *Climatic and Environmental Research*, **14**, 243–257. (in Chinese)

Storn, R., and K. Price, 1995: Differential evolution—a simple and efficient adaptive scheme for global optimization over continuous spaces. ICSI Tech. Rep. TR-95-012, 12pp.

Sun, S. F., 2005: *Biophysical and Biochemical Mechanisms and Their Parameterization in Context of Land Surface Processes.* China Meteorological Press, Beijing, 307pp. (in Chinese)

Sun, G. D., and M. Mu, 2009: A preliminary application of the differential evolution algorithm to calculate the CNOP. *Atmos. Oceanic Sci. Lett.*, **2**, 381–385.

Sun, G. D., and M. Mu, 2011: Response of a grassland ecosystem to climate change in a theoretical model. *Adv. Atmos.Sci.*, **28**, 1266–1278.

Wang, B., J. P. Wang, Z. H. Huo, P. J. Zhang, and Q. Wang, 2010: Application of conditional nonlinear optimal perturbation method in a theoretical grassland ecosystem. *Chinese Quarterly Journal of Mathematics*, **25**, 3, 422–429.

Wang, B., Z. H. Huo, P. J. Zhang, and Q. Wang, 2011a: Impacts of human activities on the transition between grassland and desert in a simplified theoretical grass field ecosystem. *Climate Change and Ecosystem Response*, 204–207.

Wang, B., P. J. Zhang, Z. H. Huo, and Q. Wang, 2011b: Application of conditional nonlinear optimal perturbation in study of lake eutrophication. *Climate Change and Ecosystem Response*, 426–429.

Xia, Y. L., A. J. Pitman, H. V. Gupta, M. Lepastrier, A. Henderson-Sellers, and L. A. Bastidas, 2002: Calibrating a land surface model of varying complexity using multi-criteria methods and the Cabauw data set. *Journal of Hydrometeorology*, **3**, 181–194.

Xia, Y. L., M. K. Sen, C. S. Jackson, and P. L. Stoffa, 2004a: Multi-dataset study of optimal parameter and uncertainty estimation of a land surface model with Bayesian stochastic inversion and multicriteria method. *J. Appl. Meteor.*, **43**, 1477–1497.

Xia, Y. L., Z. L. Yang, C. Jackson, P. L. Stoffa, and M. K. Sen, 2004b: Impacts of data length on optimal parameter and uncertainty estimation of a land surface model. *J. Geophys. Res.*, **109**, doi: 10.1029/2003JD004419.

Xie, A. H., J. W. Ren, X. Qin, and S. K. Kang, 2008: Pressure and temperature feasibility of NCEP/NCAR reanalysis sata at Mt. Everest. *Journal of Mountain Science*, **5**, 32–37.

Xie, A. H., C. D. Xiao, and J. W. Ren, 2010: Correlation between NCEP/NCAR reanalyzed air temperatures and the observed air temperatures from automatic meteorological stations along the route from Zhongshan station to dome A, East Antarctica. *Journal of Glaciology and Geocryology*, **32**, 898–905. (in Chinese)

Xin, Y. F., L. G. Bian, and X. H. Zhang, 2006: The application of CoLM to arid region of northwest China and Qinghai-Xizang Plateau. *Plateau Meteorology*, **25**, 567–574. (in Chinese)

Yu, G. Y., P. Li, Z. He, and Y. M. Sun, 2009: Advanced evolutionary algorithm used in multi-objective constrained optimization problem. *Computer Integrated Manufacturing Systems*, **15**, 1172–1178. (in Chinese)

Zheng, J., Z. H. Xie, Y. J. Dai, X. Yuan, and X. Q. Bi, 2009: Coupling of the common land model (CoLM) with the regional climate model (RegCM3) and its preliminary validation. *Chinese J. Atmos. Sci.*, **33**, 737–750. (in Chinese)

---

## Editor Comment (EC2) · O. Talagrand (Editor) · 29 Mar 2016

In view of the comments of the two referees, with which I agree, I consider the paper cannot be accepted for publication in NPG.

I thank the authors for having thought of *Nonlinear Processes in Geophysics* for submitting your paper, and I encourage them to write a new paper along the lines suggested by the referees.